# Human running performance from real-world big data

Thorsten Emig [1✉] & Jussi Peltonen [2]

Wearable exercise trackers provide data that encode information on individual running performance. These data hold great potential for enhancing our understanding of the complex interplay between training and performance. Here we demonstrate feasibility of this idea by applying a previously validated mathematical model to real-world running activities of ≈ 14,000 individuals with ≈ 1.6 million exercise sessions containing duration and distance, with a total distance of ≈ 20 million km. Our model depends on two performance parameters: an aerobic power index and an endurance index. Inclusion of endurance, which describes the decline in sustainable power over duration, offers novel insights into performance: a highly accurate race time prediction and the identification of key parameters such as the lactate threshold, commonly used in exercise physiology. Correlations between performance indices and training volume and intensity are quantified, pointing to an optimal training. Our findings hint at new ways to quantify and predict athletic performance under real-world conditions.

[1] Université Paris-Saclay, CNRS, Laboratoire de Physique Théorique et Modèles Statistiques, 91405 Orsay, France. [2] Polar Electro Oy, Professorintie 5, 90440 Kempele, Finland. ✉email: thorsten.emig@u-psud.fr

Skeletal evidence suggests that endurance running may have evolved 2 million years ago[1]. It probably originated as a hunting skill but has later developed to competition, dating back to ancient Olympic Games ~720 BC[2] and exercise form for mass population. Over the years, endurance running has undergone substantially change. Recent decades have witnessed an ever growing exercising population which uses wearable sensors to bring together astonishing volumes of data for speed, distance, heart rate, accelerations, and more[3–5]. For example, endurance athletes like runners and cyclists currently upload from GPS enabled sensors more than a billion activities per year worldwide[6]. In principle, these data provide an exciting opportunity to monitor human physiology noninvasively under real-world conditions outside the laboratory. Measuring the physiological response to physical activity can provide important insights for a variety of populations ranging from elite athletes to recreational exercisers to patients in rehabilitation[7,8]. However, the analysis of big data sets of large, heterogeneous groups of individuals poses a substantial challenge due to the quality of the data itself[9,10], lack of effective theoretical models[11], and influence of environmental factors like weather conditions[12,13]. The important, robust properties of an individual's physiology can be overshadowed by details specific to the conditions of recording. Thus, there is a demand for universal theoretical models that have been validated for noise-free exercise data and can be applied under noisy real-world conditions to derive meaningful physiological and performance information[14].

To date, exercise physiologists conventionally use laboratory testing to determine parameters that measure fitness and performance potential[15]. A strength of laboratory testing is that it can distinguish between cardiovascular limit, maximal rate of oxygen consumption ($VO_{2max}$), neuromuscular effects, and running economy[16,17]. Together VO2max and running economy determine maximal aerobic speed, which is the slowest speed at which $VO_{2max}$ occurs. Maximal aerobic speed correlates with race speed on shorter distances but alone cannot predict race times for longer distances such as the marathon. Exercise thresholds have been used in exercise testing to quantify metabolism. However, the determination of such thresholds, like the lactate threshold, in the laboratory is somewhat limited. Typical laboratory testing is short-lasting and does not always fully capture time and distance dependent reduction in running economy[18,19]. For example, only sparse results exist for the endurance limited fractional utilization of maximal aerobic power (MAP) and its dependence on exercise duration[20]. Moreover laboratory testing is expensive and not available to most of the population. The undeniable fact that the best test of running performance is an actual race and not laboratory tests highlights the need for models specifically constructed to extract performance indices of an athlete from their regular exercise performance. For these reasons, models that can utilize data from wearable devices and turn those into meaningful performance parameters may offer a cost effective alternative approach to laboratory testing. However, it must stressed that this type of approach does not elucidate the physiological and biomechanical mechanisms that control performance. It is an adjunct to the methods which are already used, providing additional insight into running and the potential training factors influencing performance and it does not replace the insights that we can gain from laboratory testing.

Several empirical and physiological models have been put forward for explaining running world records in terms of a few physiological parameters. The noted physiologist Hill empirically proposed a hyperbola to describe the maximal power output as a function of exercise duration[21]. Also a purely mechanical approach, based on the runners equation of motion, has been proposed[22]. These approaches predict that the average racing velocity tends to be a constant value with increasing race distance which contradicts observation. While more recent approaches have combined physiology and observations to propose more realistic logarithmic relations between maximal power output and duration[23], these models depend on many parameters that vary among individuals[24]. Recently we have developed a universal running model which builds on concepts in exercise physiology, depends only a minimal set of key performance indices that are required to predict race performance, contains no additional individual-dependent quantities and has been validated with running world-records[14]. Here, we show that it is also possible to obtain novel insights into individual's running performance by applying this model to big exercise datasets.

Exercise data are a valuable source of information about individual long-term training protocols. Endurance training leads to a wide spectrum of physiological responses. However, in practice, training is prescribed often only by anecdotal evidence and personal experience. This might be due to a lack of knowledge of statistically significant correlations between the relevant physiological parameters and training characteristics for large groups of individuals with different fitness status. Here, we demonstrate the feasibility to extract key performance indices from real-world running exercise data recorded with wearable exercise trackers. We apply our method to runners during their training season before a marathon race. Our universal running model characterizes a runner's performance with two indices that measure (1) endurance (endurance index) and (2) the velocity requiring MAP output (aerobic power index). The main aim of our work is to demonstrate the feasibility of extracting performance indices from real-world racing results in a big population of runners and to use these indices to predict accurate race times and evaluate the effect and efficiency of training. Our approach represents a potentially powerful platform to enlarge dramatically the number of tested subjects in sports science by extending performance index acquisition from conventional laboratory testing to real-world conditions with the aid of mathematical modeling and wearable technology.

## Results

**Universal performance model**. In previous work we have developed a model that can be used to extract aerobic performance indices from race data[14]. To summarize, this model expresses exercise intensity on a relative power scale $p$, which varies between zero, corresponding to basal metabolic rate, and unity at MAP generation. MAP is expected to correspond to maximal oxygen uptake $VO_{2,max}$ but this analogy needs not to be assumed in our approach. A linear relation $p(v)$ maps running velocity $v$ to relative power with $p(v_m) = 1$ defining $v_m$ as an aerobic power index associated with MAP beyond which anaerobic energy supply can yield $p > 1$ for a short time only. Anaerobic supply contributes to maximal exercise shorter than a crossover time $t_c$ which in our model is the longest time over which MAP can be sustained. An important prediction of our model is that the maximal value of the relative power $p$ that a runner can maintain declines logarithmically with duration, with a rate $\gamma_l$, assuming that the durations are longer than $t_c$. This finding is in agreement with a finding of A.V. Hill who observed this form of decline in running world records[21]. For more details on our model, see the "Methods" section. Here, we use this universal, i.e., subject independent model for human running performance, to extract aerobic performance indices from finishing times of runners worldwide by matching them with model predictions[14]. The analyzed data set comes from an exercise tracking platform that contains precise records of distance and duration (and hence average velocity) of running activities

of ≈19K individuals, who ran a total distance of 32M km over a period of 3.5 years. The data were recorded by the individuals with a GPS digital sports watch (V800, Polar Electro Oy, Oulu, Finland)[25], and uploaded to the platform. Maximal performance of an individual was measured by the fastest finishing time for the four most common racing distances 5000 m, 10,000 m, half-marathon (21,097.5 m) and marathon (42,195 m) within a racing season, which is defined as the 180 days preceding the marathon race (see "Methods" section for detection of racing activities).

The velocity corresponding to our parameter $v_m$ is difficult to measure in laboratory settings since $VO_{2,max}$ can be achieved over a wide range of sub-maximal intensities because of an upward drift of oxygen uptake with exercise duration[18,19]. In general, our model can determine $v_m$ from the crossover of the race–time–distance relation at time $t_c$, and hence is free from this complications. The simplest version of the model assumes a fixed time $t_c$. Model predictions for sub-MAP performances do not depend on this fixed time since other choices lead only to consistently renormalized values for $v_m$ and $\gamma_l$ (which are then no longer associated strictly with MAP but with a slightly different power). In agreement with the application of our model to running records on both the super- and sub-MAP branches[14] and laboratory testings[26], we choose $t_c = 6$ min in the following. Combining running economy and the decline of the fractional utilization of maximal power output with race duration, the fastest time $T(d)$ over a distance $d$ is given by the universal expression

$$T(d) = -\frac{t_c}{\gamma_l}\frac{d}{d_c}\frac{1}{W_{-1}\left[-\frac{d}{d_c}\frac{\exp(-1/\gamma_l)}{\gamma_l}\right]} \quad \text{for} \quad d \geq d_c , \quad (1)$$

where we defined $d_c = v_m t_c$, and $W_{-1}$ is a real branch of the Lambert $W$-function which is defined as the multi-valued inverse of the function $w \rightarrow w\exp(w)$[27]. $W_{-1}(z)$ is real valued for $-1/e \leq z < 0$ which is fulfilled for all distances $d$ that we consider (see the "Methods" section for more detail). Note that $T(d_c) = t_c$, i.e., $d_c$ is the distance that can be maximally raced in the time $t_c$. The condition $d \geq d_c$ is always satisfied for the race distances considered here. We note that Eq. (1) is an exact solution of our model. It can be also obtained from earlier descriptions of the energetics of endurance running[28–30] when the fractional utilization of MAP is described by our prediction of a slow, logarithmic decay, and a linear increase of the energy cost of running with velocity is assumed.

The model parameters, called performance indices, quantify different aspects of performance and provide a unique insight into basic determinants of fitness in a large population of runners over a wide range of exercise capacities and over long time scales. The velocity $v_m$ measures combined running economy and MAP and is known to be a better predictor of performance than $VO_{2,max}$ alone[31]. We define the endurance index as $E_l = \exp(0.1/\gamma_l)$, which encodes that 90% of $v_m$ can be maintained for an extended time $E_l t_c > t_c$. The pair of performance indices $v_m$, $E_l$ is sufficient to account for racing velocity variations for distances from $d_c$ (typically one mile in our data set) to the marathon. For example, when analyzing consistent running records of individuals, we found strong evidence that they follow the same universal scaling law of Eq. (1) as running world (or national) records do, with mean errors below 1%[14]. Here, our model estimates are based on an individual's fastest times for the four fixed racing distances, 5 k, 10 k, half-marathon, and marathon. Unfortunately, we cannot determine from the available data set if performance was achieved during an actual racing event. For our approach however, it is only required that the recorded performance

corresponds to the maximal effort over a given running distance achieved during the racing season.

**Exercise data**. An overview of the data analysis design is provided in Fig. 1. All available subjects and activities in the data set of the exercise tracking platform were grouped by SID and marathon date, combining all individual running activities during the 180 days before the marathon, defining a season. For each season, activities with the fastest time for the four fixed race distances defined a racing season. We imposed the condition that each racing season contains at least two races. If a season contained 30 or more total running activities they were defined as training season. For consistency certain data filters were applied to all activities and races (see the "Methods" section for more detail). Two variants of racing season were defined, with the marathon included and excluded. A total of ~25,000 racing seasons with the marathon included and ~10,000 racing seasons without the marathon, and ~22,000 training seasons were analyzed (see Table 1 for a summary of the available data and performed analyses).

**Accuracy of performance prediction**. For all individuals, we estimated their performance indices $v_m$ and $\gamma_l$ for each racing season by matching race events to Eq. (1) by minimizing the relative prediction error for the race times. The probability densities of these indices are shown in Fig. 2. For all racing seasons with three and more races ($N = 12,309$), the mean error between model prediction and actual race time was only 2.0%. This suggests that our model captures correctly determinants of aerobic endurance performance. Correlations between performance indices and marathon finishing times are presented in Fig. 3. To investigate the predictive power of our model in more detail, we applied our model also to the racing season with the marathon performance excluded (see Fig. 4). This allowed us to estimate the marathon finishing time from the performances on shorter distances only. As a function of performance indices, in the most likely parameter range the model predicted the marathon performance with an overall accuracy of better than 10%. Only for very small (or large) endurance $E_l$, estimated times tended to be too slow (or fast) which indicates that sub-marathon distances were raced inconsistently, leading to an under (or over) estimation of $E_l$. Given all the possible uncertainties in marathon racing that are beyond the control of this study (e.g., weather, course profile, and motivation of the athlete), our predictions for the marathon finishing times are rather satisfying.

**Maximal velocity for 1 h**. Analysis of ~25,000 racing seasons reveals a normally distributed velocity $v_m$ and an exponential decay of the probability density for the endurance $E_l$ (see Fig. 2). Interestingly, $VO_{2,max}$ in a study on 450 elite soccer players has also been found to obey a normal distribution[32]. Note that $v_m$ also measures running economy, which varies considerably among individuals and modulates performance[24]. In exercise physiology, the ability of a runner to maintain a certain effort is often characterized in terms of thresholds, of which a common example is lactate threshold. In our approach, however, there is a continuous relationship between power output and velocity, and the change of this relation with duration appears to be a natural measure for endurance capability. Hence, as a practical measure for endurance, we define in our model the velocity $v_{1hU} = v_m[1 - 0.1\log(60\,\text{min}/t_c)/\log(E_l)]$ that a runner can maintain for 1 h, corresponding to the maximal fractional utilization of MAP for 1 h. While any duration could be chosen here, we used 1 h in analogy to running coaches defining threshold velocity as the effort that can be maintained for about 1 h[33]. The 1h utilization ratio $p_{1hU} = v_{1hU}/v_m$ had been

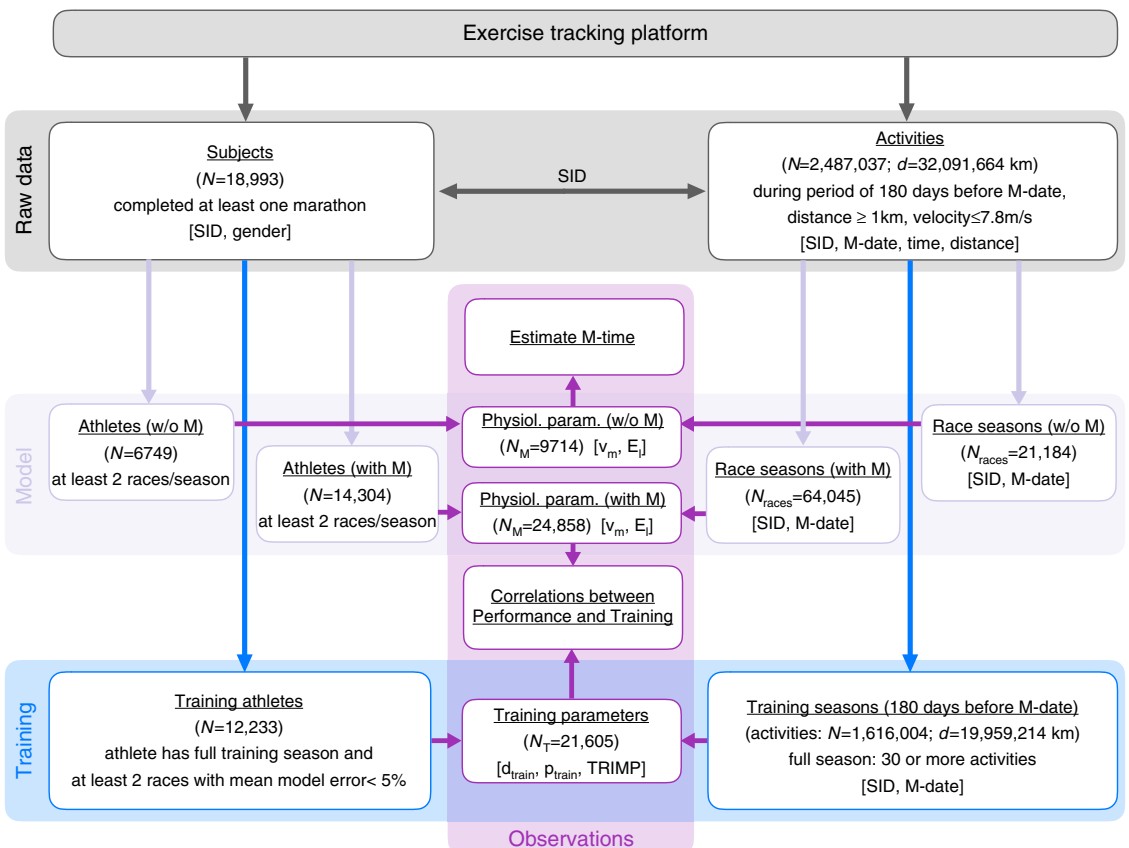

**Fig. 1 Flowchart of the exercise data analysis.** SID: subject identifier, M: marathon, M-date: date of marathon, d: total running distance, "race season": fastest times of an athlete for at least two of the distances 5 km, 10 km, half-marathon, and marathon (±3% to account for GPS tolerance), $N_{races}$: total number of races, $N_M$: number of successful model fits, $N_T$: number of analyzed training seasons for which physiological parameters $v_m$, $E_l$ could be obtained and predicted actual race times within a mean error below 5%, "full training season": at least 30 activities during the 180 days before M-date.

**Table 1 Summary of data sets analyzed.**

| Data | Available | Fit with marathon | Fit w/o marathon | Training season[c] |
|---|---|---|---|---|
| # Subjects | 18,993 | 14,304 | 6749 | 12,233 |
| # Activities[a] (distance ≥ 1 km) | 2,487,037 | | | 1,616,004 |
| Total distance [km] | 32,091,664 | | | 19,959,214 |
| Mean distance/activity [km] | 12.9 | | | 12.4 |
| # Racing events[b] | 85,993 | 64,045 | 21,184 | 54,620 |
| # Race/training seasons | | 24,858 | 9714 | 21,605 |

All data were collected through the PolarFlow web service[48].
[a]After removal of unrealistic average velocities (faster than world record).
[b]Distances are 5 km, 10 km, half-marathon, and marathon depending on the model fit (w or w/o marathon).
[c]Seasons with # runs ≥ 30.

estimated previously from laboratory measurements and races for a smaller group of 18 male long distance runners to be approximately $0.82 \pm 0.05$[34]. Strikingly, our findings from the running data for ~14,000 subjects corroborate this range without any invasive measurements, as demonstrated in Fig. 2c. Moreover, our observation of exponentially small but finite probability for larger $E_l$ explains observed values $p_{1hU} \approx 0.9$ in some well trained long distance runners.

We also computed the marathon race time from our model and compared it to the actual marathon time $T_m$ for all racing seasons, see Fig. 3. Our model predicts theoretical curves of constant $T_m$ in the plane of performance indices (shown as dashed lines in Fig. 3a). We found that the actual race times are ordered according to these curves. This shows that our selected physiological profiles, computed from sub-marathon and

marathon best performances, are highly correlated with $T_m$. It is important to understand that the position of a marathon performance in the parameter space is determined by all races and hence reflects relative importance of the indices $v_m$ and $E_l$. This demonstrates the crucial importance of taking into account endurance in addition to MAP and running economy when assessing performance of long distance runners.

**Importance of endurance.** Our findings demonstrate the strong sensitivity of performance to endurance. For example, a runner with a velocity of $v_m = 5\,\mathrm{m\,s^{-1}}$ can improve his/her marathon time from 3 h 27 min 38 s to 2 h 53 min 8 s by doubling endurance from $E_l = 3$ to $E_l = 6$ (corresponding to a change in the one-hour utilization from 79 to 87% of $VO_{2max}$), without any change in $VO_{2,max}$ or running economy. We also find that faster runners

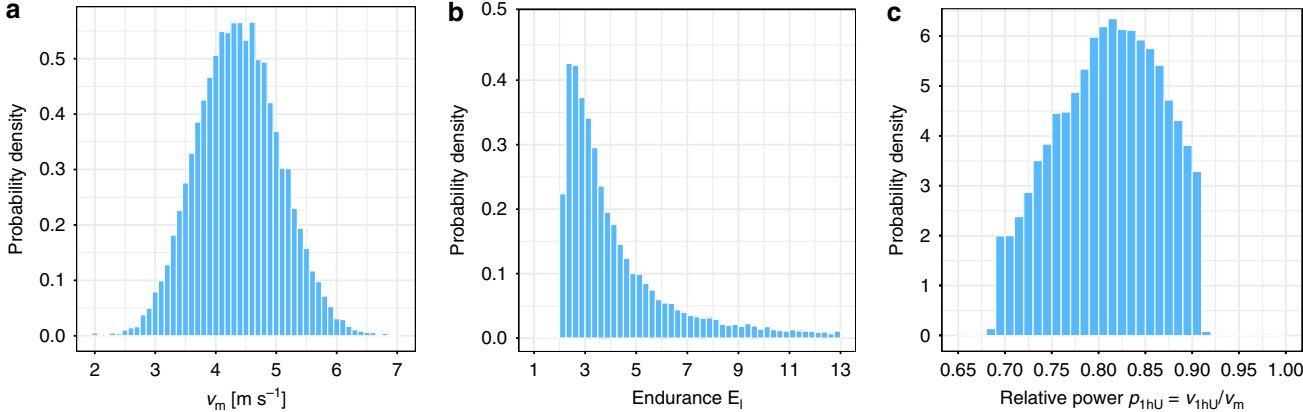

**Fig. 2 Probability density of model parameters.** The crossover velocity $v_m$ which is the smallest velocity that elicits maximal aerobic power MAP and the endurance $E_l$ are obtained by applying our model to the fastest performances of a subject for the four distances 5 km, 10 km, Halfmarathon, and Marathon of a racing season. For these distributions, a total of 24,858 racing seasons have been analyzed. **a** The velocity $v_m$ is approximately normally distributed with a mean of 4.4 m s$^{-1}$. **b** The probability density for the endurance $E_l$ resembles an exponential decay. **c** The probability density for the relative power for 1h utilization (1hU) peaks at about 82% of MAP.

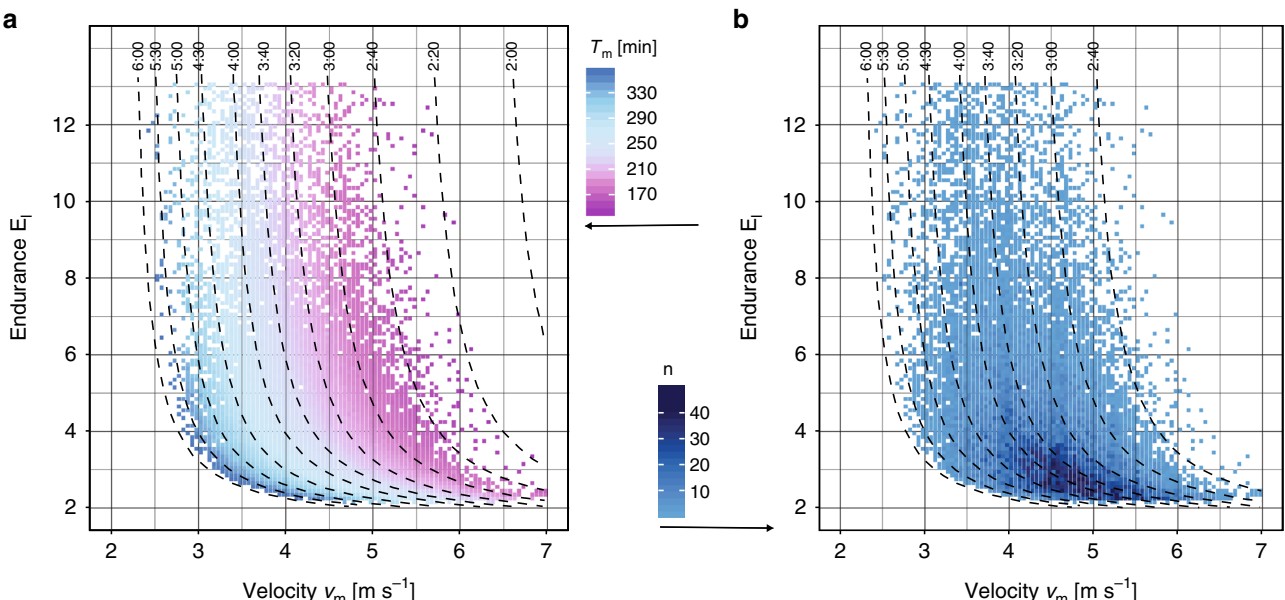

**Fig. 3 Correlation between performance indices and marathon race time (model estimates for 24,504 racing seasons are shown here). a** Visualization of the marathon race time $T_m$ in the ($v_m$, $E_l$) parameter plane. Performance indices are obtained from individual's best performances during the racing season. Color changes from fast (magenta) to slower (blue) finishing times (see color legend for time in minutes). Parameter pairs ($v_m$, $E_l$) along the dashed curves yield the same marathon race time indicated at the top of the graph (in hh:mm format). **b** Color coded visualization of the number $n$ of racing seasons analyzed as function of the parameters ($v_m$, $E_l$).

tend to race more consistently over all race distances than slower runners, highlighted by the dependence of the prediction error $\Delta T_m$ on the marathon finishing time (see Fig. 4b). For example, within our fastest group of runners with a marathon time below 160 min, the prediction error was typically less than ±2.5%. This observation supports our explanation for the observed uncertainty in the endurance parameter $E_l$.

**Correlation with training.** Finally, we compared physiological profiles to running activities within a training season. There exist a few studies of the relation between training volume and intensity, improvements of aerobic fitness and performance[35]. For example, it has been stated that running at velocity $v_m$ might represent an optimal stimulus for improving endurance[36]. There is also evidence supporting that a relatively large percentage of

low-intensity training over a long period improves performance during highly intense endurance events[37,38]. It has been argued that running velocity at lactate threshold is the best physiological predictor for distance running performance[39].

To investigate the effect of training distance and speed, relative to the velocity $v_m$, we selected consistent racing seasons defined by having a mean race time prediction error below 5%. Figure 5a shows that as the total training distance $d_{train}$ of the training season increases, $v_m$ increases on average linearly, with a weak saturation trend at largest $d_{train}$. Several studies have demonstrated an increased $v_m$ due to endurance training[35]. A faster velocity $v_m$ can be achieved by a better running economy and/or an increase in MAP. We hypothesize that longer training distance has generated improved running economy, in agreement with earlier observations in a group of eleven well-trained long

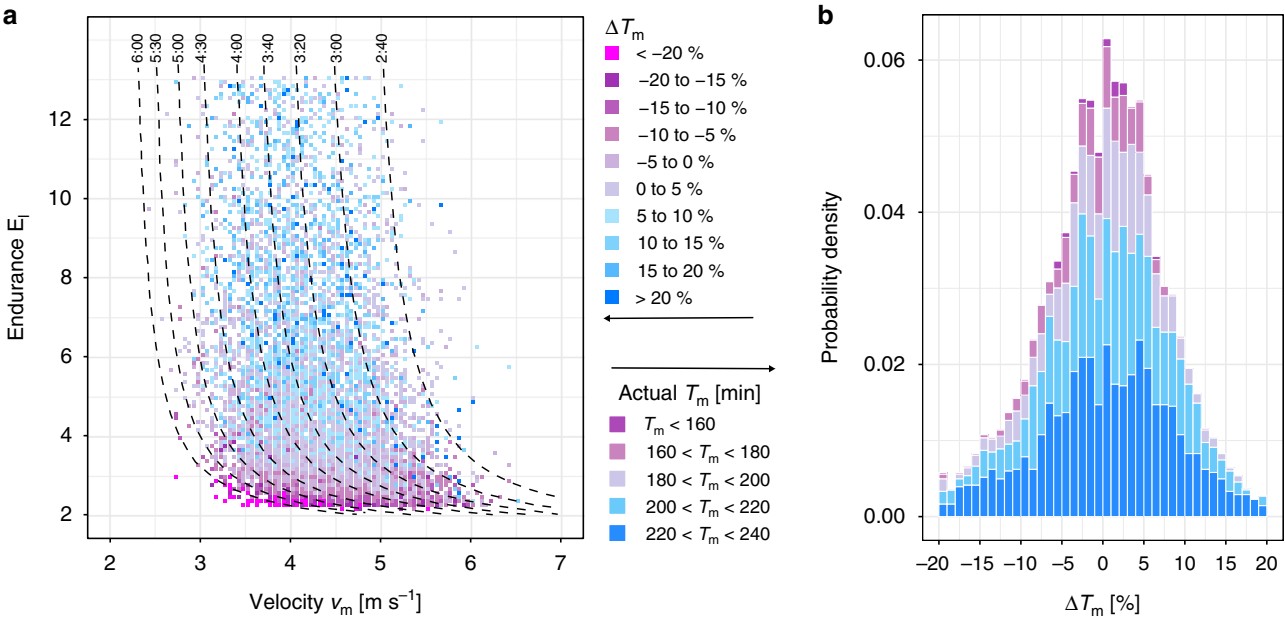

**Fig. 4 Estimate of Marathon race time from the racing season (for 9410 seasons). a** Visualization of the relative difference $\Delta T_m$ between actual and estimated marathon race time $T_m$ (in percent of race time) as function of crossover velocity $v_m$ and endurance $E_l$. Magenta (blue) color indicates a faster (slower) than estimated finish. **b** Probability density of race time differences color coded according to groups of different race time intervals.

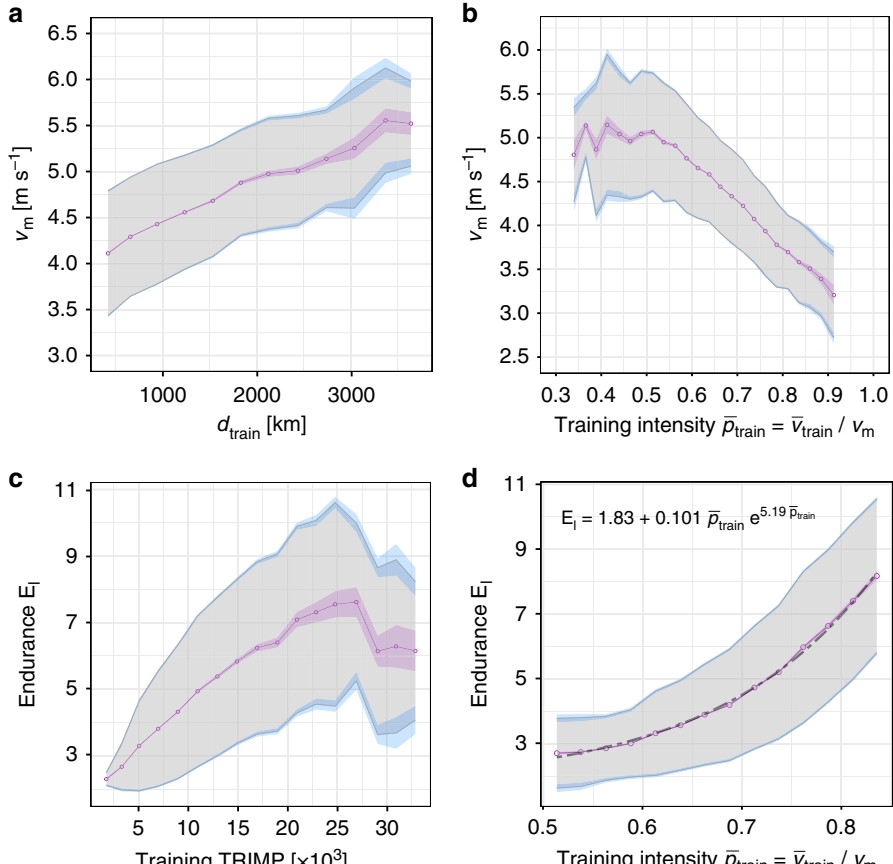

**Fig. 5 Correlations between performance indices and training characteristics.** We have measured the distance and time for each running activity during a training season for a total number of 21,605 seasons. The graphs show the observed relations between performance indices (obtained from a model fit to the racing season) and different measures of training volume and intensity. The magenta line indicates the average, the gray region one standard deviation, and the light magenta and blue shaded areas represent the standard error of the mean and standard deviation, respectively, as obtained from bootstrap resampling with replacement (see "Methods" section for more details). **a** Increase of crossover velocity $v_m$ with total distance $d_{train}$ of training runs. **b** Relation between crossover velocity $v_m$ and relative training intensity $p_{train} = \bar{v}_{train}/v_m$ where $\bar{v}_{train}$ is the average training velocity. **c** Increase and saturation of endurance $E_l$ with training impulse (TRIMP). **d** Exponential growth of endurance $E_l$ with relative training intensity.

distance runners[40]. Our analysis provides a statistically significant, quantitative relation between training distance and speed at MAP, $v_m$, for ~22,000 training seasons. Another explanation for this relation could be that fitter runners with a larger MAP and hence higher $v_m$ log more kilometer during their training. Unfortunately, we could not measure $v_m$ at the beginning and the end of the training season independently from two different racing seasons or time trials. We also found a linear decrease of $v_m$ with the mean relative training intensity between 50% and about 90% of $v_m$, as shown in Fig. 5b. Our findings can be interpreted as faster runners train typically at lower relative intensities which is consistent with high-intensity performance improvement due to low-intensity training. The range of training velocities increases with larger $v_m$ which reflects a wider range of accessible intensities between minimal (jogging) and maximal speed. For example, a runner with $v_m = 4\,\mathrm{m\,s^{-1}}$ typically (within one standard deviation) trains between 64 and 84% of $v_m$ or MAP, while a runner with $v_m = 5\,\mathrm{m\,s^{-1}}$ trains typically up to 66% of $v_m$ so that both runners have an almost identical upper pace ~5 min km$^{-1}$ for the majority of their runs. Slow runners must train at a relative high intensity if they want to avoid a transition to walking. It is important to realize that these typical ranges do not include fast, high-intensity workouts which account only for a small fraction of total training volume. However, high-intensity sessions involve also resting phases that can reduce the average velocity when timer is not stopped, potentially explaining observed intensities below ~50% of $v_m$.

**Optimal training impulse**. We found strong evidence that combined effect of training volume and intensity, known as TRaining IMPulse (TRIMP)[41], enhances endurance only up to a limit. Previously, it was found in recreational long distance runners that individual TRIMP correlates with 5000 m and 10,000 m track performances[42]. We computed TRIMP by summing the TRIMP points of all runs of the training season. For each run, TRIMP points were assigned according to the duration of the run and its relative average velocity $\bar{v}/v_m$ (see "Methods" section for details). We analyzed the quantitative relation between endurance $E_l$ and total TRIMP of a training season (see Fig. 5c). We observed an initial linear increase of $E_l$ with TRIMP, a plateau around $E_l = 7.5 \pm 2$ for TRIMP ~25,000, and a statistically significant final drop which may be due to over-training. This result suggests that there is an optimal TRIMP per training season, and the corresponding maximal endurance enables a close to optimal marathon race time for a given velocity $v_m$ (see Fig. 3a). Finally, we probed the definition of TRIMP itself to determine if it implements the best relation between endurance and training intensity. We found a striking agreement between the exponential dependence of $E_l$ on $\bar{v}_{train}/v_m$ and the original definition of TRIMP based on the rise of blood lactate with intensity, as demonstrated in Fig. 5d. Our findings for thousands of runners show that relations between training mode and performance indices that are usually only accessible by invasive and resource-consuming laboratory testing can be obtained reliably from running activity data.

## Discussion

Recent advances in wearable sensor technology have enabled real-time and noninvasive measurement of physiological data during exercise. However, if we are to employ these data to better understand interplay between exercise, performance and human health, we must develop new models that are adapted to extract from the raw data quantities that are most relevant for health and performance assessment. In this work we have taken this approach for long distance running to estimate physiological

model indices such as MAP and endurance, and examined their correlations with training volume and intensity by analyzing exercise data of ~14,000 marathon runners worldwide. We found that our recent universal model for a logarithmic relation between fractional utilization of maximal power and exercise duration[14] is crucial for going beyond previous approaches which ignored this relation, and for defining a parameter measuring endurance. This is an important complement to physiological testing in the laboratory where the required maximal effort is unpractical to achieve for distances over 20 km. Indeed, our results provide evidence of the possibility to extract precise indicators for performance and fitness status from long-duration real-world exercise tracking data. Using automated digital exercise tracking goes beyond previous outside-lab studies that relied often on frequently inaccurate self-reports of exercise. The probability distributions of the extracted performance indices show large variances, implying that studies with only a few individuals might produce misleading results, missing the large interindividual variability of response to exercise.

Our work has also some limitations: For each activity, only total distance and duration was available in the data set. This could lead to biased estimates of the mean velocity, for example due to periods of rest or stopping with the device timer not stopped. For the detected correlations between performance indices and training the direction of any cause–effect relationship remained open: for example, training with a higher total TRIMP might produce better endurance, but higher endurance could also enable runners to follow training modes with a higher TRIMP. To resolve this relationship, additional data filters need to be developed to select groups of runners with similar initial performance which subsequently follow different training modes. However, the observed correlations can be of practical importance. They can be useful for estimating realistic expectations for a race for less experienced runners from their training intensity and volume. In addition, our observation that endurance peaks at a given training load (TRIMP) should help preventing over-training, i.e., unproductive increase in training that can cause injury and other health problems. It should also be stressed that real-world data always lack the controlled environment of laboratory based testing. For example, the energy cost of running has been measured very accurately in laboratory conditions[43–46] and the theoretical approaches derived from these experiments have motivated the development of our model.

Our work implies several directions for future research. The combination of effective models and real-world exercise data holds great potential for a change in our theoretical description and understanding of human response to physical activity over longer periods of time, optimal exercise dosing and training, early injury detection and prevention, and elite athlete performance. Approaches similar to ours could be used to develop standards for cardiorespiratory fitness based on the probability distribution of performance indices in populations with certain characteristics. More detailed, time-resolved activity data for heart rate, mechanical power output and others could be integrated in our model to improve accuracy and to extract other performance indices. Further applications of our approach include the detection of the usage of performance enhancers in professional sports, the early identification of talented athletes, and even the effect of sports equipment like new running shoe technology on performance indices[47].

## Methods

**Exercise tracking platform**. Exercise data were obtained from Polar Flow web service[48], which is an exercise tracking platform that allows users to upload various exercise data, including running distance and velocity from GPS watches. Meta data and activity data of users are linked anonymously through user identification.

**Selection of subjects and activities**. Users of the exercise tracking platform were selected as subjects for this study under the conditions that they had completed a run over the marathon distance (42,195 m) in the period between 1 Jul 2015 and 31 Dec 2018, and used the same GPS watch (Polar V800) for activity recording to assure comparable accuracy of GPS based distance recording. We analyzed the running data of ~19,000 individuals who completed ~2.5M activities with a total distance of ~32M km (see Table 1 for details). For each individual all running activities in the 180 days before a completed marathon race were grouped together with the marathon race and the groups labeled uniquely by a subject identifier (SID) and the marathon date (M-date). Note that an individual may have have completed multiple marathons during the studied period. For each of those groups, labeled by the pair (SID, M-date), a race season was defined as the fastest runs of all activities over the four race distances 5km, 10km, half-marathon (21,097.5m) and marathon (42,195 m), if distances were available. A tolerance of ±3% was allowed in the distance selection to account for GPS inaccuracy, and average race velocities were determined by assuming the actual race distances (which are more reliable than GPS recordings). We applied conditions that race velocities must increase with decreasing race distance and must be slower than current world record velocities. Inconsistent race seasons were identified by violation of these conditions and excluded from further analysis. Race seasons were defined both with and without the marathon race included. A valid race season must contain at least two different race distances. For each race season with a successful performance model fit with mean race time error below 5% (see section below) a corresponding training season was defined as all running activities with a total distance ≥1000 m in the 180 days before the marathon. Runs with apparent velocities ≥7.8 m s$^{-1}$ (world record for 1000 m) were excluded. Only training seasons with 30 or more runs were considered so that runner had trained at least once per week and training seasons with longer interruptions were excluded.

**Performance model**. We mathematically describe running performance by a minimal model based on a relative power scale[14]. The model is formulated in terms of relative quantities to eliminate irrelevant, subject dependent quantities. The nominal power expenditure $P(v)$ that is required to run at a constant velocity $v$, the so-called running economy, determines the relative power as

$$p(v) = \frac{P(v) - P_b}{P_m - P_b} = \frac{v}{v_m} , \quad (2)$$

where we introduced a basal power $P_b$ that is obtained by linearly extrapolating the running economy to zero velocity and a crossover power $P_m$ that we expect to be close to the MAP associated with maximal oxygen uptake VO$_{2max}$. This power $P_m$ defines a crossover velocity $v_m$ that is close to the velocity that permits exercise with maximal time at MAP. For velocities $v > v_m$ the energy cost of running cannot be determined from oxygen uptake alone due to anaerobic energy supply.

The running performance of an athlete is not only determined by $p(v)$ (which is fixed by running economy and VO$_{2max}$) but depends crucially on the average power $P_{max}$ that can be maximally generated over a duration $T$ over which it can be sustained. To run at the average velocity $v_{max}$ that can be maximally sustained over the time $T$, the nominal power $P(v_{max}) = P_{max}(T)$ is required, establishing a relation between $v_{max}$ and $T$. It has been shown[14] that $P_{max}(T)$ can be obtained from a self-consistency relation which states that the time average of the instantaneously utilized power $P_{max}(T - t)$ equals the sum of $P_{max}(T)$ and a supplemental power. This supplemental power has aerobic and anaerobic contributions and accounts for an upward shift in the power that is required to complete a run with a given average velocity, for example, due to deteriorating running economy or muscle fatigue. The existence of an upward shift has been observed experimentally and it is essential since its absence would yield a duration independent $P_{max}$, which contradicts the fact that a given power cannot be sustained for an arbitrary duration. The solution of the self-consistency equation yields

$$P_{max}(T) = P_m - P_l \log \frac{T}{t_c} \quad \text{for} \quad T \geq t_c , \quad (3)$$

where $P_l$ measures the supplemental power supply and $t_c$ is a crossover time scale separating different anaerobic and aerobic forms of supplemental power. It can be shown that for $T < t_c$, $P_{max}$ is given by Eq. (3) with $P_l$ replaced by another constant. By inverting $P_{max}(T)$ and using the power–velocity relation of Eq. (2), we get the maximal time $T_{max}(v) = t_c \exp[(v_m - v)/(\gamma_l v_m)]$ over which an average velocity $v$ can be sustained. Here, the constant $\gamma_l = P_l/(P_m - P_b)$ measures endurance $E_l = \exp(0.1/\gamma_l)$, see main text. The shortest time $T(d)$ for covering a distance $d$ follows from solving $T = T_{max}(v = d/T)$ for $T$, yielding Eq. (1). It is important for the application to a large, inhomogeneous group of subjects that this model is universal in the sense that it only depends on three parameters $v_m$, $t_c$, and $\gamma_l$ and does not depend directly on any additional, subject-dependent parameters.

**Performance data analysis**. We tested whether or not meaningful performance indices can be deduced only from the racing performance of individuals, employing the performance model described before. For each racing season, uniquely labeled by a pair (SID, M-date), two model parameters, $v_m$ and $\gamma_l$, were computed from Eq. (1) applied to all races in the racing season. In general, the time $t_c$ must be obtained from the crossover between anaerobic and aerobic regimes, and hence from races that involve both means of energy supply, i.e., events with finishing time

shorter and longer than $t_c$. Explicit comparison to racing results and laboratory testing has shown that $t_c = 6$ min is a good approximation on average, and this estimate was used in our data analysis[14]. We numerically minimized the sum of the squared relative differences between the actual race time and the one predicted by Eq. (1). The nonlinear fitting was based on a Levenberg–Marquardt type algorithm with multiple starting values to minimize probability to converge only to local minimum, and with support for lower and upper parameter bounds. Parameter bounds were chosen as 2 m s$^{-1}$ ≤ $v_m$ ≤ 7 m s$^{-1}$, 0.039 ≤ $\gamma_l$ ≤ 0.135 corresponding to 2.1 ≤ $E_l$ ≤ 13.0[14]. Fits that converged onto these bounds were excluded from further analysis.

**Training data analysis**. To quantify training of individuals during the 180-day period before a marathon, we must establish measures based on duration and distances of activities within the training season. We considered an optimal set of three variables that measure quantity, quality, and a combination of quantity and quality. Training volume was quantified by total running distance $d_{train}$ of a training season. To account for possibly varying physiological adaptions during different training modes, training intensity $p_{train} = \bar{v}_{train}/v_m$ was measured by the average running velocity $\bar{v}_{train}$ in relation to the characteristic velocity $v_m$ that was determined for each race season independently. Finally, the overall training load was evaluated by the TRIMP scale, which is frequently employed in exercise physiology and the design of training. TRIMP is a measure for both volume and intensity of exercise. We assigned to each activity of a training season a TRIMP number using the definition TRIMP $= T_{train} \kappa_1 (\bar{v}/v_m) \exp(\kappa_2 \bar{v}/v_m)$ for activity of duration $T_{train}$ and average velocity $\bar{v}$ with $\kappa_1 = 0.64$, $\kappa_2 = 1.92$ for male subjects, and $\kappa_1 = 0.86$, $\kappa_2 = 1.67$ for female subjects[49]. The total training TRIMP number was then obtained by summing the individual TRIMP numbers of all activities within a training season. Usually TRIMP is defined in terms of the average heart rate reserve during exercise which is expected to be well approximated by the ratio $\bar{v}/v_m$. We are interested in the relation between physiological model parameters $v_m$ and $E_l$, and training variables. To measure these relations, we grouped training variables into bins of widths $\Delta d_{train} = 300$ km, $\Delta p_{train} = 0.025$ and $\Delta$TRIMP $= 2000$. The standard error of the mean and of the standard deviation of $v_m$ and $E_l$ within each bin was estimated by bootstrap resampling with replacement and computation of the standard deviation from 1000 bootstrap replicates.

**Reporting summary**. Further information on research design is available in the Nature Research Reporting Summary linked to this article.

## Data availability
The data that support the findings of this study are available from Polar Electro Oy but restrictions apply to the availability of these data, which were used under the license for the current study, and so are not publicly available. Data are, however, available from the authors upon reasonable request and with permission of Polar Electro Oy (research@polar.com).

## Code availability
The code (R-script) is available from the Zenodo website https://doi.org/10.5281/zenodo.4008806.

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

## Acknowledgements

The support by Polar Electro in obtaining the exercise data from their data base is greatly acknowledged.

## Author contributions

T.E. designed the study and performed the numerical analysis. T.E. and J.P. wrote the paper.

## Competing interests

The authors declare no competing interests.
