## [Peer Review File · Nature Communications]

Reviewers' Comments:

Reviewer #1:

Remarks to the Author:

The authors are correct in identifying that an explosion of data collection from wearable exercise apps has the potential to enable new insight into exercise bioenergetics and fatigue. However, while I like the general approach taken by the authors and appreciate the labour involved in their study, I believe they have 'missed a trick' in limiting their analysis to the 'universal model of running performance' described by Mulligan et al in 2018 (and not yet validated by others). The 'critical power' model is well established in the field and has both theoretical and practical validity but is unfortunately not used by the authors; indeed, the authors are somewhat (and unjustifiably, in my opinion) dismissive of the CP concept. In summary, as presented, the study is limited to a novel and essentially unvalidated model of running performance and this calls into question at least some of the conclusions. While the approach (data mining) surely has merit, the analyses need to be less blinkered and more comprehensive in this first step.

Reviewer #2:

Remarks to the Author:

In this study, the researchers show that a model they previously developed to predict future race performance from past race performances performs well in a large dataset of race times and distances estimated from a running watch. They extract parameters from the model related to (1) endurance and (2) running speed and VO₂ Max and show that there are associations between these two parameters and training metrics extracted from the dataset.

The dataset is exciting, the associations with training are interesting, and as far as I can tell, the analysis as performed was sound overall. But there are some issues with the manuscript as written and the analysis as performed that limit impact. Of most significance is that the novelty is not overwhelmingly clear. For example, the model has been previously published and shown to relate well to real-world performance data, so these aspects of the current paper are not particularly novel. The physiological parameters are shown to vary among the population, but this is not particularly novel besides the means in which the parameters were extracted. The correlations with training performance are just correlations and, as the authors acknowledge, it is not possible to determine whether the training measures associated with higher performance cause those higher performances or are merely associated with being a high-performing athlete.

Given the size and, I expect, richness of the dataset, I imagine that there is much more that the investigators could have learned. I list a few questions in the following paragraphs that the authors might have explored.

Do aspects of training help to explain errors in the model predictions? This could have helped us move toward a more causal link between training and race performance. There also seemed to be a systematic error in their model related to the endurance parameter. Is this related to training? Or errors in their data? Or a gap in their model? This question should have been explored more fully.

Are there means to predict an athlete's race performance from sub-maximal training performance (i.e., not races), using heart rate or any other measures the watch might provide? The current model requires subjects to performance two or more races at maximal effort to extract these parameters. While this is an improvement on physiological testing it is still a burden and does not seem to take advantage of the dataset. I presume heart rate is available, for example. Would heart rate and heart-rate variability during training help to detect some of the physiological parameters on training runs?

Do the measures extracted from their model and the real-world dataset match measures extracted from gold standard lab assessments in a small (but heterogeneous) subset of the subjects? While the researchers do compare to previously published data, these tests would have provided more convincing evidence that their model is valid in a population with varying age, gender, ethnicity, and training status.

Does their phenomenological model perform better than past models on this large dataset? What about a simple linear regression model? Comparing the models against additional baselines would have provided further confidence.

Another major contribution could be to share the dataset with other researchers, which would be highly novel and a means to accelerate research on human performance, injury, and real-world training. I would not expect the researchers to tackle all of these problems, but I would expect more novel insights or contributions in some form.

Another issue with the current submission is that I found the manuscript more challenging to read and understand than needed. Work is needed to improve the readability and clarity of the writing. As one example, the abstract as written contains very few specific details about the study that was performed. What parameters were predicted? What is performance? What are training modes? Given space, the abstract should not be exhaustive, but the key details should be described with enough specificity to give the reader a more clear understanding of what the study entailed. The Introduction, Results, and Discussion need similar improvements to more clearly and succinctly state what analysis was performed.

Reviewer #3:

Remarks to the Author:

MAJOR

In the Introduction, the authors challenge an axiom that has been characterizing exercise physiology since longer than a century, the axiom that measurement conditions should be standardized. I kindly disagree with this view. Existing models, validated experimentally in the laboratory, and applicable on the field and on large-scale numbers, come from standard experimental laboratory conditions. The theoretical models do exist indeed. They have been developed theoretically and validated by measuring $\dot{V}O_2\text{max}$ during exercise testing and the fraction of $\dot{V}O_2\text{max}$ utilization and the energy cost of the locomotion mode at stake at steady state. The basic formula is as follows:

$$V = f * \dot{V}O_2\text{max}/C \quad (1)$$

Where v is velocity, f the sustainable fraction of $\dot{V}O_2\text{max}$ over a given distance, and C the energy cost of the locomotion mode at stake. C has the dimension of a force and represents the metabolic energy that is to be spent to generate the force that is necessary to overcome the forces that oppose to the body movement. These are two forces: 1) air resistance (or water resistance in swimming) and 2) frictional forces. These can be expressed as follows:

$$C = k v^2 + C_f \quad (2)$$

This equation means that, if you plot C as a function of the square of speed, you obtain linear relationships with slope equal to constant k and y -intercept equal to the energy cost that is necessary to overcome frictional forces (high in running, low in cycling). Constant k is directly proportional to the frontal surface area of the moving body, to the aerodynamic (or hydrodynamic factor) C_x , and to the air (or water) density. With these simple relationships, it is possible to simulate and interpret all what happens in field conditions and during actual competitions. I just give the authors a few references (clearly overlooked by the authors) to make them aware of what I am saying: di Prampero PE, Int J Sports Med 7: 55-72, 1986; di Prampero PE, Eur J Appl Physiol 82: 345-360, 2000; di Prampero et al, J Appl Physiol 47: 201-206, 1979; di Prampero et al, Eur J Appl Physiol 55: 259-266, 1986; Ferretti et al, Eur J Appl Physiol 111: 391-401, 2011; Margaria et al, J Appl Physiol 18: 367-370, 1963; Minetti et al, J Appl Physiol 93: 1039-1046, 2002;

Pendergast et al, J Appl Physiol 43: 475-479, 1977; Tam et al, Eur J Appl Physiol 112: 3797-3806, 2012; Zamparo et al, Eur J Appl Physiol 111: 367-378, 2011. All the elements that are necessary to set a model to be applied to large data sets and in field conditions are present in those and many other studies.

What is the use of a variable like running economy, which the authors define as the steady state oxygen consumption at a given constant speed instead of C ? Physically speaking, C looks more appropriate.

At the end of page 2, the authors create artificially a dichotomy between laboratory tests and actual competition conditions: nevertheless, the theoretical background derived from laboratory tests is fully applicable to laboratory conditions.

A sentence like "Unfortunately, these approaches predict that speeds below a critical velocity can be maintained for infinite duration which contradicts observation" has been criticized (see e.g. Ferretti, Energetics of Muscular Exercise, Springer 2015, chapter on critical power). Laboratory physiologists are perfectly aware of the fact that energy sources in the body are finite.

Under Results, I read: "Main determinants of aerobic fitness and endurance in long distance runners (LDR) are maximal oxygen uptake per body weight, VO_{2max} , velocity dependent, sub-maximal oxygen demand, known as running economy (RE), and the lactate or ventilatory threshold (LT) that sets the limit below which a steady state blood lactate concentration is maintained". Although a reference is given, this statement is not correct, see equation 1 in this report.

Equation 1 of the article is constructed in such a way to encompass a large variety of field conditions, independent of the physiological model. Frankly, I do not see a role for such a study, unless we create an artificial opposition between classical physiological studies and a kind of "modern" approach that the authors claim. Big numbers are fascinating, but their utility depend of the context into which they are inserted and interpreted. The context exists, but the authors seem to be unaware of it.

The discussion is biased by the chosen approach and does not deserve to be discussed analytically
MINOR

Endurance running dates back much longer than the ancient Olympic Games: it is alike that pre-historical nomadic societies used endurance running while hunting or for migrating.

Page 2, line 7 : ranging instead of raging

Rebuttal letter**“Novel insights on human exercise performance from big data mining”
(NCOMMS-20-02292)**

Please find below our point-to-point answers to the reviewer comments (C: comment, A: answer).

Answer to Reviewer #1

We thank the reviewer for her/his time spent looking over our manuscript and their comments that we address point-by-point in the following.

C *The authors are correct in identifying that an explosion of data collection from wearable exercise apps has the potential to enable new insight into exercise bioenergetics and fatigue. However, while I like the general approach taken by the authors and appreciate the labour involved in their study, I believe they have ‘missed a trick’ in limiting their analysis to the ‘universal model of running performance’ described by Mulligan et al in 2018 (and not yet validated by others). The ‘critical power’ model is well established in the field and has both theoretical and practical validity but is unfortunately not used by the authors; indeed, the authors are somewhat (and unjustifiably, in my opinion) dismissive of the CP concept.*

A We acknowledge that the ‘critical power’ model is well established. A. V. Hill described the idea behind this concept already in 1925 (*The Physiological Basis of Athletic Records, Lancet, 1925*). He derived the idea from running and other world records as he noted that maximum speed or power over time T follows a hyperbolic curve that can be described by the equation $P_{\max}(T) = P_c + A/T$, where P_c corresponds to critical power (although Hill did not use that term) and A represent anaerobic power reserve. Furthermore, Hill noted that this relation is *limited to durations up to 12 minutes* and called it short-term fatigue. He thought that short-term fatigue originates from muscles whereas other forms of fatigue that take longer to develop have more complex origins, such as neural fatigue, and are therefore much harder to describe. Thus, his idea of ‘critical power’ was not meant to describe human performance over *long-duration exercise*. Currently, the ‘critical power’ model is mainly applied to running distances up to 10km. For longer distances, the concept of a duration dependent *fractional utilization* of maximal aerobic power is required, as pointed out also by Reviewer #3 (see also work by di Prampero, and Peronnet & Thibault). In fact, Hill indicated in Figure 4 of his 1925 paper that the average running velocity tends to decrease *logarithmically* with race duration (we have attached this figure as an appendix to this rebuttal). Our universal model for running performance, as described in our previous paper in 2018 and used in the present work, builds on Hills observation. It describes running performance over a much broader exercise duration band. However, we do acknowledge that critical power has been useful in some applications. For example, it has been used in cycling, where loading of the muscles and fatigue is different from running. Below, we have attached a table in which we summarize the ‘critical power’ and other models. In fact, the ‘critical velocity’ v_c (corresponding to critical power) indeed occurs in our model (as described in Mulligan et al in 2018) as a combination of parameters, i.e., $v_c = v_m - D'/t_c$ where we followed the notation of the ‘critical power’ model used in A. M. Jones, A. Vanhatalo, *Sports Med* 47, S65 (2017). In order to highlight the difference between the ‘critical power’ model and our model, we have performed a detailed comparison of the models, using running world records from 1987 and 2020, and also personal records from six elite marathon runners taken from the above mentioned publication of A. M. Jones & A. Vanhatalo. Corresponding results are attached below. As the relation between the models is not directly relevant to our present data analysis of race distances between 5km and the Marathon, we have removed the reference to ‘critical power’ to avoid any confusion and to not give a false impression of this concept.

C *In summary, as presented, the study is limited to a novel and essentially unvalidated model of running performance and this calls into question at least some of the conclusions. While the approach (data mining) surely*

has merit, the analyses need to be less blinkered and more comprehensive in this first step.

A While we agree that our model is novel, we disagree with the conclusion that it is "essentially unvalidated". As pointed out by Reviewer #2, *"the model has been previously published and shown to relate well to real-world performance data"*. We note that other researchers have checked their models also by comparison to athletic records for a certain range of distances, and so did we for our model. To provide a constructive basis for further review, and to clear up misunderstandings, but also to defend our mathematical model used for analyzing the data, we provide below a new, rather detailed comparison of the 'critical power' model and our model, including new graphs and tables. Our new results show that our model agrees with current athletic world and personal records with an error of less than 1%. We are not aware of any mathematical model that explains current world records from 800m to the Marathon at better accuracy.

Rebuttal letter**“Novel insights on human exercise performance from big data mining”
(NCOMMS-20-02292)**

Please find below our point-to-point answers to the reviewer comments (C: comment, A: answer).

Answer to Reviewer #2

We thank the reviewer for her/his time spent looking over our manuscript and their comments and interesting questions that we address point-by-point in the following.

C *The dataset is exciting, the associations with training are interesting, and as far as I can tell, the analysis as performed was sound overall. But there are some issues with the manuscript as written and the analysis as performed that limit impact. Of most significance is that the novelty is not overwhelmingly clear. For example, the model has been previously published and shown to relate well to real-world performance data, so these aspects of the current paper are not particularly novel. The physiological parameters are shown to vary among the population, but this is not particularly novel besides the means in which the parameters were extracted. The correlations with training performance are just correlations and, as the authors acknowledge, it is not possible to determine whether the training measures associated with higher performance cause those higher performances or are merely associated with being a high-performing athlete.*

A Novelty of the current paper is that it can explain running performance from 5km up to the marathon in a large group of runners, with a wide range of performance levels, using two effective parameters: the crossover velocity v_m and the endurance parameter E_l . Usually, running performance is measured by VO₂max alone, which is a poor indicator of performance as it ignores running economy (the energy cost of running per distance) which shows a considerable variation among athletes. In addition, running economy changes (slowly) over time, which is believed to be associated with various forms of fatigue and change of physiological parameters like, e.g., body temperature. Given the complexity of these mechanisms and their current poor understanding, it appears interesting that two parameters are rather effective in describing running performances of a few hours duration. It should be stressed that our model is the first to model running over these long time scales by a *logarithmic decay* in fractional utilization (FU) of maximal aerobic power. Previous models considered constant or linear decrease in FU, leading to systematic errors for distances over 10km (Only the model by Peronnet & Thibault considers a combination of logarithmic and power law decays). We have added in the appendix to this rebuttal a detailed comparison of our model to other existing models that were suggested by the other reviewers.

In our work, we think for the first time, one can see how training history is associated with key performance parameters on a large population level. It should be noted that endurance is *impossible* to measure in the laboratory as it would require multiple hours of running on a treadmill which presumably involves a number of artificial effects compared to 'real world' running. Hence, there is currently no reliable evaluation of correlations between real-world running endurance and training beyond some studies of individual athletes. It is, however, correct that our analysis does not determine if training is the cause of observed performance, and just associated with higher performance level. While this is an interesting question for future analysis, even the here detected correlations can be of practical importance: They can be useful for estimating realistic expectations for a race for less experienced runners from their training intensity and volume, and hence prevent "hitting the wall" early in the race. In addition, our observation that endurance peaks at a given training load (in TRIMP), see Fig. 5(c), should help preventing over-training, i.e., unproductive increase in training that can cause injury and other health problems.

C *Given the size and, I expect, richness of the dataset, I imagine that there is much more that the investigators could have learned. I list a few questions in the following paragraphs that the authors might have explored.*

Do aspects of training help to explain errors in the model predictions? This could have helped us move toward a more causal link between training and race performance. There also seemed to be a systematic error in their model related to the endurance parameter. Is this related to training? Or errors in their data? Or a gap in their model? This question should have been explored more fully.

A We feel that we should first clarify the content of our data set since it is less rich as your comments suggest. Our model contains only the date, total distance and average velocity for all runs of the subjects. (See below for a comment on heart rate.) While more data are recorded by GPS watches, these time series with one second resolution could not be provided by our industrial partner for millions of kilometers of running. However, we agree that separate work on a smaller number of subjects with higher data detail would be very interesting and should be performed in the future.

Regarding observed deviations between actual race times and model predictions, we first note that our model has been shown to outperform all existing models when applied to personal records from 800m to the marathon of elite runners (please see appendix to this rebuttal). The systematic error in the predicted marathon times in Fig.4(a) at rather small and large endurance appear to be a consequence of the difficulty to measure endurance from a few races at shorter distances when these races are not performed under perfect conditions or optimal motivation of the athlete. As Fig. 4(b) shows, this problem is most pronounced for slower runners. Fast runners demonstrated the smallest error between prediction and actual race time. This is consistent with the observation that fast runners display also highest consistency in performance over all race distances (due to higher experience and more racing attempts on a given distance), and hence their endurance parameter shows less uncertainty. This is particularly the case for elite runners (see analysis in the appendix). As there were associations between performance indicators and training background (Fig. 5), we can draw a similar conclusion: Low relative training intensity and high training volume, typical for more experienced and faster runners, is associated with smaller model error.

C *Are there means to predict an athletes race performance from sub-maximal training performance (i.e., not races), using heart rate or any other measures the watch might provide? The current model requires subjects to performance two or more races at maximal effort to extract these parameters. While this is an improvement on physiological testing it is still a burden and does not seem to take advantage of the dataset. I presume heart rate is available, for example. Would heart rate and heart-rate variability during training help to detect some of the physiological parameters on training runs?*

A Estimating race performance from sub-maximal training performance directly is impossible without additional assumptions being made. An important quantity for endurance running performance is the decay of fractional utilization of maximal aerobic power with duration which measures for how long an runner can maintain a certain fraction of maximal aerobic power output. This quantity can be estimated from 'time to exhaustion' experiments in the laboratory, i.e., by maximal tests. Without a precise knowledge of this quantity (measured by E_t in our model), a 'typical' value can only be assumed (depending on training status). Our dataset contains for most athletes a number of races over 5km to the halfmarathon as these distances are used during training as 'test races'. Hence, for marathon runners, this information on maximal effort events is usually available and provides a clear improvement over physiological testing in the laboratory where maximal effort is impossible to motivate for a distance of 20km or longer.

As far as heart rate is concerned, our data set does not contain heart rate data for all runs and athletes as not all runners who wore a GPS watch wear a heart rate monitor (chest strap). But even if this data would be available, there remains an important unknown: the maximal heart rate of the athlete which varies substantially among

individuals and cannot be determined accurately and easily from age-based formulas. Without the maximal heart rate the important *relative* effort (the quantity p in our model) cannot be determined accurately. Because of this, our running model was built on the requirement that a priori no information about the runner's physiology is needed. Additional challenges are that heart rate is affected by external factors, such as temperature, that are often unknown. The same goes for heart rate variability as even less is understood about how different levels of heart rate variability during exercise relate to effort or athletic performance.

C *Do the measures extracted from their model and the real-world dataset match measures extracted from gold standard lab assessments in a small (but heterogeneous) subset of the subjects? While the researchers do compare to previously published data, these tests would have provided more convincing evidence that their model is valid in a population with varying age, gender, ethnicity, and training status.*

A When it comes to endurance running tests conducted in the laboratory, there are several reasons why they should not be considered as benchmark for running performance: (1) Maximal tests in laboratory are difficult to repeat, possibly due to lack of motivation to go all-out without opponent or competition or even prize money to win. As a result, the coefficient of variation may be as high as 25% (Billat et al., *Med Sci Sports Exerc*, 1994; Wigley et al., *Int J Sports Med*, 2007); (2) Running mechanics varies considerably between treadmill and over ground running (Nigg et al., *Med Sci Sports Exerc*, 1994; Sinclair et al., *Sport Biomech*, 2012). One reason may be difficulty to simulate wind resistance; (3) Maximal laboratory tests are short-lasting and therefore fail to account for reduction in running economy and subsequent increase in oxygen consumption at given speed that occurs over long-distance running. We note also that all existing models for running performance have been validated by comparison to athletic records, and not by laboratory testing. Demographic and other measures that rely on user input are not reliable in big data sets from tracking platforms as ours as many users never update default settings. Only location, which is given by GPS, is considered reliable.

C *Does their phenomenological model perform better than past models on this large dataset? What about a simple linear regression model? Comparing the models against additional baselines would have provided further confidence.*

A The most realistic test of models is their agreement with running world records and personal records of elite athletes since those data are most consistent and obviously obtained under maximal effort and controlled settings. A variety of models have been proposed in the past. Only one of them, proposed by Peronnet & Thibault [F. Peronnet, G. Thibault, *Mathematical analysis of running performance and world running records*, *J Appl Physiol.* 67, 453 (1989)] employs a *logarithmically* decaying fractional utilization of maximal aerobic power, based on empirical observations in athletic performances. Their model predicts world-records with an error of less than 1% but the model is complicated by the fact that it requires many physiological parameters (body weight, running economy, etc) that are unrealistically assumed to be the same for every athlete. While our model is similar to the one by Peronnet & Thibault it is different in two essential points: (1) The logarithmic decay of fractional utilization of maximal aerobic power emerges in our model from an exact solution of a self-consistency equation and (2) our model is universal in the sense that it depends only on relative (rescaled) quantities and hence can be applied to all athletes without knowing details like, e.g., body weight and size, and running economy. Our model predicts world records from 800m to the marathon with an error slightly less than the one observed for the model of Peronnet & Thibault. To provide a constructive basis for further review, and to clear up misunderstandings indicated by the other reviewers, but also to defend our mathematical model used for analyzing the data, we provide below a new, rather detailed comparison of the Peronnet & Thibault model and some other models (mentioned by the other reviewers) and our model, including new graphs and tables. The attached tables and graphs also show that a linear regression would not work since the race velocities change on a logarithmic time scale, with a marked crossover at about 2000m race distance. While we think that our new comparison can help the evaluation of our present work, it would not improve the manuscript

but would only make it more exhaustive to read. We note that details of our model and its validation against world records have been published earlier [M. Mulligan, G. Adam, T. Emig, *A minimal power model for human running performance*, PLoS ONE 13(11): e0206645 (2018)].

C *Another major contribution could be to share the dataset with other researchers, which would be highly novel and a means to accelerate research on human performance, injury, and real-world training. I would not expect the researchers to tackle all of these problems, but I would expect more novel insights or contributions in some form.*

A Following general policy, our data set shall be made available to other researchers upon request once our work has been published.

C *Another issue with the current submission is that I found the manuscript more challenging to read and understand than needed. Work is needed to improve the readability and clarity of the writing. As one example, the abstract as written contains very few specific details about the study that was performed. What parameters were predicted? What is performance? What are training modes? Given space, the abstract should not be exhaustive, but the key details should be described with enough specificity to give the reader a more clear understanding of what the study entailed. The Introduction, Results, and Discussion need similar improvements to more clearly and succinctly state what analysis was performed.*

A We have rewritten some parts of the manuscript to improve clarity of the description of performed analysis. Specifically, the abstract contains now more details about our study.

Rebuttal letter**“Novel insights on human exercise performance from big data mining”
(NCOMMS-20-02292)**

Please find below our point-to-point answers to the reviewer comments (C: comment, A: answer).

Answer to Reviewer #3

We thank the reviewer for her/his time spent looking over our manuscript and their comments that we address point-by-point in the following.

C *In the Introduction, the authors challenge an axiom that has been characterizing exercise physiology since longer than a century, the axiom that measurement conditions should be standardized. I kindly disagree with this view. Existing models, validated experimentally in the laboratory, and applicable on the field and on large-scale numbers, come from standard experimental laboratory conditions. The theoretical models do exist indeed. They have been developed theoretically and validated by measuring VO₂max during exercise testing and the fraction of VO₂max utilization and the energy cost of the locomotion mode at stake at steady state. The basic formula is as follows:*

$$V = F * \dot{V}O_{2max} / C \quad (.1)$$

where v is velocity, F the sustainable fraction of VO₂max over a given distance, and C the energy cost of the locomotion mode at stake. C has the dimension of a force and represents the metabolic energy that is to be spent to generate the force that is necessary to overcome the forces that oppose to the body movement. These are two forces: 1) air resistance (or water resistance in swimming) and 2) frictional forces. These can be expressed as follows:

$$C = kv^2 + C_f \quad (.2)$$

This equation means that, if you plot C as a function of the square of speed, you obtain linear relationships with slope equal to constant k and y-intercept equal to the energy cost that is necessary to overcome frictional forces (high in running, low in cycling). Constant k is directly proportional to the frontal surface area of the moving body, to the aerodynamic (or hydrodynamic factor) C_x, and to the air (or water) density. With these simple relationships, it is possible to simulate and interpret all what happens in field conditions and during actual competitions. I just give the authors a few references (clearly overlooked by the authors) to make them aware of what I am saying: di Prampero PE, Int J Sports Med 7: 55-72, 1986; di Prampero PE, Eur J Appl Physiol 82: 345-360, 2000; di Prampero et al, J Appl Physiol 47: 201-206, 1979; di Prampero et al, Eur J Appl Physiol 55: 259-266, 1986; Ferretti et al, Eur J Appl Physiol 111: 391-401, 2011; Margaria et al, J Appl Physiol 18: 367-370, 1963; Minetti et al, J Appl Physiol 93: 1039-1046, 2002; Pendergast et al, J Appl Physiol 43: 475-479, 1977; Tam et al, Eur J Appl Physiol 112: 3797-3806, 2012; Zamparo et al, Eur J Appl Physiol 111: 367-378, 2011. All the elements that are necessary to set a model to be applied to large data sets and in field conditions are present in those and many other studies.

A We thank the Reviewer for discussing the details of the model developed by P. .E. di Prampero et al. These remarks suggest that there has been a misunderstanding which could be due to our very brief discussion of our model and in particular its relation to other models. Let us hence clarify this point, by using your notation for the model. *Our model is exactly equivalent to above equations (.1) and (.2) with a particular form for the sustainable fraction F and a constant, velocity independent C which has been used previously by others [see, e.g., S. Lazzar et al., Eur J Appl Physiol, 112, 1709 (2012)] and is justified for the running velocities in our data*

set (negligible air resistance). While the reviewer does not provide an explicit expression for F , this model has been applied to half and full marathon races, using for the sustainable fraction

$$F(T) = f_0 - f_1 T, \quad (.3)$$

i.e., a linearly decreasing function of the duration T of the race, with constants f_0, f_1 [P. E. di Prampero et al., Eur J Appl Physiol 55, 259 (1986)]. A related model has been developed by Peronnet & Thibault [F. Peronnet, G. Thibault, J Appl Physiol, 67, 453 (1989)], using a logarithmic function for the sustainable fraction,

$$F(T) = 1 + \frac{E}{\text{MAP}} \log(T/T_{\text{MAP}}), \quad (.4)$$

with maximal aerobic power MAP, a negative constant E and $T_{\text{MAP}} = 7\text{min}$. Peronnet & Thibault motivated this choice by empirical arguments based on world record performances up to the Marathon distance. Interestingly, we have shown in our paper in 2018 [M. Mulligan, G. Adam, T. Emig, PLoS ONE 13(11): e0206645 (2018)] that the form of Eq. (.4) can be derived mathematically from a self-consistent integral equation. In the notation of our present manuscript, the sustainable fraction is given by

$$F(T) = \frac{P_{\text{max}}(T)}{P_m} = 1 - \frac{P_l}{P_m} \log \frac{T}{t_c} \quad (T > t_c), \quad (.5)$$

with $P_m = \text{MAP}$ and $t_c = 6\text{min}$, see Eq. (3) of our manuscript. Substituting this equation in your Eq. (.1) yields exactly our model. This and additional details of the relation between the model you described above, the so-called 'critical power' model proposed by another Reviewer, and the model by Peronnet & Thibault are summarized in the attached table. We have also performed a new, extensive comparison of these models to current running world records and personal records of some elite marathon runners, including the model proposed in above Eqs. (.1) and (.2) with F given by Eq. (.3). All results are attached below. They show that our model has overall the smallest average error for the considered athletics records. We note that the 'critical power' model and above model with F given by Eq. (.3) show substantial discrepancies with running records for distances longer than $\sim 10\text{km}$ (see attached plots). Hence, we believe that (1) a logarithmic decay of F is essential and (2) our model is a very reasonable approach to analyze the race distances of 5km, 10km, Halfmarathon and Marathon in our data set.

C *What is the use of a variable like running economy, which the authors define as the steady state oxygen consumption at a given constant speed instead of C ? Physically speaking, C looks more appropriate.*

A We agree that the energy cost of running, C , is the appropriate quantity. In fact, as pointed out in the previous item, our model does employ this concept. The exact relation between C in your equation and our model is

$$v * C = P_b + \frac{P_m - P_b}{v_m} v \quad (.6)$$

where $P_m = \text{MAP}$ and P_b is the resting (basal) metabolic rate (power). This relation means that we measure the energy cost of running in our model by a parameter v_m which is a velocity that is close to the running speed that can be maintained for about 6min, equivalent to the time scale T_{MAP} in the model of Peronnet & Thibault. Please note that this implies the relation $C_f = (P_m - P_b)/v_m$ and $v_m * C = \text{MAP}$.

C *At the end of page 2, the authors create artificially a dichotomy between laboratory tests and actual competition conditions: nevertheless, the theoretical background derived from laboratory tests is fully applicable to laboratory conditions.*

A It is not our intention to suggest a general discrepancy between laboratory testing and actual race performance. Our explanations on the items above show that we indeed use the theoretical background that you suggest. The

crucial difference between previous approaches and ours is based on our result for the duration dependence of the sustainable fraction F . And hence the point we want to rise here is the often relative short duration of laboratory testing. Incremental running test is the most common laboratory test that is conducted to determine aerobic and anaerobic thresholds as well as maximal aerobic speed and maximal heart rate. However, incremental running test is short-lasting and cannot account for the effect of exercise duration on thresholds or general effects of fatigue. The maximal fractional utilization $F(T)$ can be investigated in time-to-exhaustion test such as running at certain fraction of VO_{2max} , but the obtained results may have low test-retest repeatability as indicated by a 25% coefficient of variation [Billat et. al., Med Sci Sports Exerc, 1994; Wigley et al., Int J Sports Med, 2007]. Furthermore, running mechanics between treadmill and over ground running are different [Nigg et al., Med Sci Sports Exerc, 1994; Sinclair et al., Sport Biomech, 2012]. In conclusion, laboratory tests are most suitable for observing changes in running performance over relative short durations, but test results may not always accurately predict actual race performance due to a lack of knowledge of the function $F(T)$ and also due to differences in running mechanics that occur between treadmill and outdoor ground. Also, an important aspect when comparing laboratory testing and actual races in which world records are set is the degree of motivation of the athlete. This latter point seems particularly relevant to long lasting time-to-exhaustion tests performed to determine F .

- C *A sentence like Unfortunately, these approaches predict that speeds below a critical velocity can be maintained for infinite duration which contradicts observation has been criticized (see e.g. Ferretti, Energetics of Muscular Exercise, Springer 2015, chapter on critical power). Laboratory physiologists are perfectly aware of the fact that energy sources in the body are finite.*
- A With this statement on the 'critical power' model we wanted to point out the importance of using a fractional utilization $F(T) < 1$ of maximal aerobic power when describing long lasting events like the marathon.
- C *Under Results, I read: Main determinants of aerobic fitness and endurance in long distance runners (LDR) are maximal oxygen uptake per body weight, VO_{2max} , velocity dependent, sub-maximal oxygen demand, known as running economy (RE), and the lactate or ventilatory threshold (LT) that sets the limit below which a steady state blood lactate concentration is maintained. Although a reference is given, this statement is not correct, see equation 1 in this report.*
- A As explained before, the difference between equation (1) in your report and our model consists in the function used to describe the fractional utilization $F(T)$. The "lactate or ventilatory threshold (LT)" is defined in our article from the duration dependence of $F(T)$ as the fractional utilization of MAP that the runner can maintain for one hour. We have changed the name and description of this threshold in our article accordingly to avoid confusion with other concepts such as LT. As also explained before, the energy cost of running is measured in our model by the velocity v_m which is directly related to C in equation (1) in this report.
- C *Equation 1 of the article is constructed in such a way to encompass a large variety of field conditions, independent of the physiological model. Frankly, I do not see a role for such a study, unless we create an artificial opposition between classical physiological studies and a kind of modern approach that the authors claim. Big numbers are fascinating, but their utility depend of the context into which they are inserted and interpreted. The context exists, but the authors seem to be unaware of it. The discussion id biased by the chosen approach and does not deserve to be discussed analytically.*
- A Equation 1 of our article is in fact an exact mathematical solution of the eq. (1) given in this report, with the fractional utilization of MAP given by $F(T) = 1 - \log(T/t_c)$ for a race of duration $T > t_c$ with the time scale $t_c = 6\text{min}$ in this article. This form for $F(T)$ was derived in our earlier work [M. Mulligan, G. Adam, T. Emig, PLoS ONE 13(11): e0206645 (2018)]. Hence, our chosen approach fits fully into the existing context after the importance of $F(T)$ is understood.

C *Endurance running dates back much longer than the ancient Olympic Games: it is alike that pre-historical nomadic societies used endurance running while hunting or for migrating.*

A Thank you for this interesting remark. We have modified the beginning of the introduction to give a more general presentation.

C *Page 2, line 7 : ranging instead of raging*

A Thank you. We corrected this spelling error.

Appendix to Rebuttal Letter: New results from a comparison of existing mathematical models

A. V. Hill: The physiological basis of athletic records (1925)

In his seminal work, Hill posed the question "how long a given effort can be maintained". To answer this question he analyzed running records. In Figure 4 of his original article (reproduced in Fig. 1 below) he plotted the average running speed over the a logarithmic time scale. It can be seen that for running (and other sports) the velocity decays linearly with the logarithm of time, following two branches with different slopes. The analysis of Peronnet & Thibault [F. Peronnet, G. Thibault, *Mathematical analysis of running performance and world running records*, J Appl Physiol. 67, 453 (1989)] and our mathematical model, applied to current world records, confirm Hill's observation with high accuracy, as shown in the next section. This logarithmic decay is not reproduced by the models proposed by Reviewers #1 and #3.

FIG. 4.

Records for men skating, bicycling, running, and walking, and for women running. Horizontally, logarithm of time occupied in race; vertically, average speed in yards per second. The same scale is used throughout, except for bicycling, where half the scale is employed, as shown in square brackets. The curve for men running appears to be somewhat doubtful beyond 10 or 15 miles, and three alternative curves are shown by broken lines.

FIG. 1 Original figure from A. V. Hill, *The physiological basis of athletic records*, The Lancet, September 5, 1925, showing average speed for running and other sports over a logarithmic time scale.

Comparison of mathematical models

The mathematical models for running performance mentioned by the reviewers (reviewer #1: critical power model, reviewer #3: di Prampero's approach) and the model by Peronnet & Thibault are summarized and compared in Table I. The last column of this table provides the relation of those models to our model.

In order to assess and compare the accuracy of these models and our model we have performed detailed analyses of men running world records (1987 as in Peronnet & Thibault, and current as of April 2020) and personal records of six elite marathon runners (Antonio Pinto, Eliud Kipchoge, Felix Limo, Haile Gebrselassie, Mo Farah, Steve Jones; choice of athletes taken from A. M. Jones, A. Vanhatalo, Sports Med 47, S65 (2017); Data from <https://www.worldathletics.org/athletes>). For all models the unknown parameters were determined by minimizing the mean squared relative error between the theoretically predicted time and the actual race time, i.e., the expression

$$\text{Err} = \frac{1}{N} \sum_{j=1}^N \left(\frac{T_{\text{theory}}(d_j) - T_{\text{race}}(d_j)}{T_{\text{race}}(d_j)} \right)^2 \quad (.7)$$

was minimized where the sum extends over N race distances d_j . A numerical algorithm based on differential evolution was used for this purpose. The following models were analyzed:

MIT: Our model, here called the 'MIT model'

[M. Mulligan, G. Adam, T. Emig, PLoS ONE 13(11): e0206645 (2018)]

CP: The 'critical power' model

[see e.g. M. Jones, A. Vanhatalo, Sports Med 47, S65 (2017)]

PT: The model of Peronnet & Thibault

[F. Peronnet, G. Thibault, J Appl Physiol. 67, 453 (1989)]

diP: The model of di Prampero (with $F(T)$ given by Eq. (.3) with $f_0 = 1$)

[see e.g. P. E. di Prampero et al., Eur J Appl Phys 55, 259 1986]

Following the analysis in A. M. Jones, A. Vanhatalo, Sports Med 47, S65 (2017), for the CP model the race distances were restricted to $d_j < 15.000\text{m}$ for the determination of the model parameters.

The analyzed race distances and times are listed along with the obtained model parameters in the attached tables, see Figs. 2 and 4. Shown are also the errors of the model predictions for each race distance and the average error (av.error) for each model. The attached plots in Figs. 3 and 5 show the race results (open circles) and the four model predictions for the average race velocity $\bar{v}(d)$ as function of the race distance d as solid curves. The velocity is measured in units of v_m and the distance in units of $d_c = v_m t_c$ which corresponds to a simple linear rescaling of time and distance.

We decided to plot average velocity (in units of the velocity v_m at maximal aerobic power, MAP) since this shows clearly the relative slow decay of velocity with racing distance. For example, the world records show that a marathon is raced just $\sim 18\%$ below the velocity at MAP. This means that a mathematical model needs to achieve a rather high precision in predicting the mean velocity in order to properly distinguish between endurance running distances.

Summary of results from analyzing running records

Our findings are as follows:

1. For all analyzed data sets, the average error between the model prediction and the actual race times is smallest for the MIT model, followed by the PT model. It should be noted that both models *describe the fractional utilization*

of maximal aerobic power by a logarithmic function. The typical error of both models for the marathon is well below 1%, and with the average error of our MIT model being less than half of the error of the PT model for world records.

2. The CP model shows a *systematic discrepancy for distances over 10km and below 1500m*. The predicted average velocity tends to a constant ("critical velocity") with increasing distance, indicated by a dashed line in the plots. The typical error for the marathon varies around 8%, both for world and personal records.
3. The diP model also shows a *systematic error in the range of long race distances*. The curve for the mean velocity shows a non-monotonous curvature that bends towards too small velocities for larger distances, leading to a typical error of a few percent for the half and full marathon.
4. Interestingly, for most data sets (in particular the world records), three model predictions converge (intersect) on one particular point that is defined in the MIT model by the velocity v_m and the distance $d_c = v_m t_c$, corresponding approximately to the time scale $T_{\text{MAP}} \sim t_c$ in the PT model over which the velocity $v_m \sim v_{\text{MAP}}$ can be maintained. This observation has important consequences: It shows that all four models tend to agree with increasing accuracy when the velocity $v_m \sim v_{\text{MAP}}$ is approached. This implies that 'critical power' or 'critical velocity' can be obtained from the MIT model. This is indeed the case, and the relation is summarized in the attached Tab. I.
5. The data from world records are described by all models in general better than personal records of individual athletes since world records are a result of optimized preparation and talent of an athlete for a given distance. However, even on the level of individual athletes, the MIT model outperforms the other models, as shown by the modeling of elite marathon runners (see Tab. 4 and Fig. 5).

We conclude that the models based on a constant or polynomial function $F(T)$ for the fractional utilization of MAP give an approximate description of running records that is valid only for distances below the 5km or the 10km race or durations below 15 to 30min. This time scale is consistent with the 15min duration already observed by A. V. Hill in his 1925 paper for the ending of a rapid decrease of race velocity and the beginning of a slower, logarithmic fall. Indeed, for larger distances, a logarithmic function $F(T)$, as used in the PT model and the our model, is essential for a consistent description of real world running records.

TABLE I Summary of performance models.

model, reference	main variables and equations	relation to our model ('MIT model')
CRITICAL POWER (CP) Monod & Scherrer (1965)	The model is expected to describe races from 800m up to 5km or perhaps 10km. Power $P(v)$ and velocity $v(T)$ sustainable over time T: [A. M. Jones, A. Vanhatalo, Sports Med 47, S65 (2017)] $P(T) = P_c + \frac{W'}{T}, \quad \text{or} \quad v(T) = v_c + \frac{D'}{T}$ with critical power P_c and critical speed (CS) v_c, anaerobic capacity W' (in W/kg) or distance D' (in m). Fractional utilization is fixed at unity: Power $P < P_c$ or velocity $v < v_c$ can be maintained for "infinite" time but in praxis limited by substrate.	CP model close to our model around duration t_c with relation $v_c \approx v_m - \frac{D'}{t_c}$ and $D' \approx \frac{P_s}{P_m + P_s} v_m t_c$ No description of fractional utilization of MAP, corresponding to $P_l = 0$ in our model.
DI PRAMPERO (DIP) di Prampero (1986)	Maximal velocity $v(T)$ sustainable for time T [di Prampero et al., J Appl Physiol 74,2318 (1993)]: $v(T) = \frac{F(T)}{C(v(T))} \dot{E}_{\max}, \quad \dot{E}_{\max} = \frac{A}{T} + \text{MAP} - \text{MAP} \frac{\tau}{T} (1 - e^{-T/\tau})$ with work A from anaerobic sources, maximal aerobic power MAP, $\tau = 10$s, and the energy cost of running [per distance and body weight in J/(m kg)] given by $C(v) = C_f + kv^2 + 2v^3/d \quad (v \text{ in m/s, } d \text{ in m})$ with $k = 0.0103$, $C_f = 3.79$. Fractional utilization $F(T)$ of MAP over duration T is approximated by $F(T) = f_0 - f_1 T$ where $f_0 \approx 1$, $f_1 \approx 0$ for $T < 20$min, and $f_0 \approx 0.94$, $f_1 \approx 10^{-3}$ for durations from a half to a full marathon with T in min. [di Prampero et al., Eur J Appl Phys 55, 259 1986].	MAP $\hat{=} P_m$ Fractional utilization of MAP: $F(T) = 1 - \frac{P_l}{P_m} \log \frac{T}{t_c} \quad (T > t_c)$ Power output required to run at velocity v: $C(v)v = P_b + \frac{P_m - P_b}{v_m} v,$ so that $C_f = (P_m - P_b)/v_m$ and $k = 0$. This means $C(v_m)v_m = P_m$ implying that v_m is speed at MAP.
PERONNET & THIBAUT (PT) Peronnet & Thibault (1989)	Power output $P(T)$ sustainable over time T and power $P_v(v)$ required to at velocity v: $P(T) = \begin{cases} c_2(T)^{\frac{A}{T}} + \text{MAP} - c_1(T)(\text{MAP} - \text{BMR}) & (T < T_{\text{MAP}}) \\ c_2(T)^{\frac{A}{T}} \left(1 + f \log \frac{T}{T_{\text{MAP}}}\right) + c_1(T)\text{BMR} + (1 - c_1(T)) \left(\text{MAP} + E \log \frac{T}{T_{\text{MAP}}}\right) & (T > T_{\text{MAP}}) \end{cases}$ $P_v(v) = \text{BMR} + 3.86v + C'v^3 \quad (v = d/T \text{ in m/s})$ with $c_2(T) = 1 - e^{-T/k_2}, \quad c_1(T) = \frac{k_1}{T} \left(1 - e^{-T/k_1}\right).$ with maximal aerobic power MAP (in W/kg), anaerobic capacity A in J/kg, $T_{\text{MAP}} = 7$min the maximal race duration for which the peak aerobic power is MAP, rate of peak decline E in W/kg, $k_1 = 30$s, $k_2 = 20$s, $f = -0.233$, $C' = 0.0103 + 2/d$ with distance d in m and basal metabolic rate $\text{BMR} = 1.2 \text{W/kg}$	MAP $\hat{=} P_m$, BMR $\hat{=} P_b$, $T_{\text{MAP}} = t_c$ $A, f, C' = 0$ Our model does not include kinetics of aerobic and anaerobic metabolism at the beginning of exercise (< 30s) so that $c_1(T) = 0$, $c_2(T) = 1$. Fractional utilization of MAP is the same as in our model, i.e., logarithmic decrease with $E = -P_l$ and for $T < T_{\text{MAP}}$, A/T is replaced by $-P_b \log \frac{T}{t_c}$ with $A \approx P_b t_c$.

ID	distance	time	MIT	error[%]	CP	error[%]	PT	error[%]	dip	error[%]	MIT parameters	CP parameters	PT parameters	dip parameters	
WR1987men	800	01:41.73	01:42.12	+0.39	01:32.50	-9.07	01:41.67	-0.06	01:40.19	-1.51	t_c 05:28.54		V_{MAP} 6.36 m/s	V_{MAP} 5.96 m/s	
	1000	02:12.18	02:11.57	-0.46	02:05.83	-4.80	02:12.60	+0.31	02:11.69	-0.37	v_{in} 6.76 m/s		E/MAp -5.00 %	f_i $1.85 \times 10^{-5}/s$	
	1500	03:29.46	03:29.09	-0.18	03:29.16	-0.14	03:30.23	+0.37	03:31.63	+1.03	$E_a = T_{110\%MAP}/t_c$ 0.480		A	A	
	1609	03:46.32	03:46.62	+0.13	03:47.33	+0.45	03:47.18	+0.38	03:49.18	+1.26	$E_a = T_{90\%MAP}/t_c$ 5.493				
	2000	04:50.81	04:51.15	+0.12	04:52.50	+0.58	04:48.07	-0.94	04:52.34	+0.53	Anaerobic & aerobic metabolism				
	3000	07:32.10	07:32.41	+0.07	07:39.16	+1.56	07:25.84	-1.38	07:34.92	+0.62	$A = P_{1c} + (P_{m1} - P_{b1})25s$ 1820.0 J/kg	D'			
	5000	12:58.39	12:59.38	+0.13	13:12.49	+1.81	13:04.46	+0.78	13:03.24	+0.62	$D' = v_{in} t_c (P_{1c} + P_{m1})$ 255.7 m	CS			
	10 000	27:13.81	27:13.51	-0.02	27:05.82	-0.49	27:32.87	+1.17	27:00.38	-0.82	$CS = v_{in} - D'/t_c$ 5.98 m/s				
	21 100	1:00:55.00	1:00:35.14	-0.54	57:55.81	-4.90	1:00:49.44	-0.15	59:29.22	-2.35	$VO2max$ 78.4 ml/(kg min)				
	42 195	2:07:12.00	2:07:39.60	+0.36	1:56:31.62	-8.39	2:06:33.66	-0.50	2:08:44.79	+1.22	av.error 0.24 %	av.error 3.22 %	av.error 0.60 %	av.error 1.03 %	
	WR2020men	800	01:40.91	01:41.71	+0.79	01:33.77	-7.07	01:41.32	+0.41	01:40.20	-0.70	t_c 05:14.60		V_{MAP} 6.54 m/s	V_{MAP} 6.19 m/s
		1000	02:11.96	02:10.58	-1.04	02:05.89	-4.60	02:11.49	-0.36	02:10.77	-0.90	v_{in} 6.93 m/s		E/MAp -4.48 %	f_i $1.77 \times 10^{-5}/s$
		1500	03:26.00	03:26.09	+0.04	03:26.18	+0.09	03:27.05	+0.51	03:28.12	+1.03	$E_a = T_{110\%MAP}/t_c$ 0.435			
		1609	03:43.13	03:43.08	-0.02	03:43.68	+0.25	03:43.53	+0.18	03:45.08	+0.87	$E_a = T_{90\%MAP}/t_c$ 6.732			
2000		04:44.79	04:45.40	+0.21	04:46.47	+0.59	04:42.74	-0.72	04:46.09	+0.46	Anaerobic & aerobic metabolism				
3000		07:20.67	07:20.91	+0.05	07:27.05	+1.45	07:15.31	-1.22	07:22.99	+0.53	$A = P_{1c} + (P_{m1} - P_{b1})25s$ 1678.0 J/kg	D'			
5000		12:37.35	12:36.66	-0.09	12:48.20	+1.43	12:41.29	+0.52	12:39.51	+0.29	$D' = v_{in} t_c (P_{1c} + P_{m1})$ 224.4 m	CS			
10 000		26:17.53	26:16.99	-0.03	26:11.10	-0.41	26:33.79	+1.03	26:05.02	-0.79	$CS = v_{in} - D'/t_c$ 6.21 m/s				
21 100		58:01.00	58:05.83	+0.14	55:53.52	-3.66	58:17.54	+0.48	57:11.12	-1.43	$VO2max$ 80.3 ml/(kg min)				
42 195		2:01:39.00	2:01:34.01	-0.07	1:52:20.93	-7.65	2:00:36.40	-0.86	2:02:35.30	+0.77	av.error 0.25 %	av.error 2.72 %	av.error 0.63 %	av.error 0.78 %	

FIG. 2 Application of four mathematical models to men running world records from 1987 and 2020: Predicted race times and model parameters (see Tab.I for models).

distance	time	CP model	error [%]	PT model	error [%]	dIP model	error [%]	MIT model	error [%]
800	01:41.73	01:32.50	-9.07	01:41.67	-0.06	01:40.19	-1.51	01:42.12	+0.39
1000	02:12.18	02:05.83	-4.80	02:12.60	+0.31	02:11.69	-0.37	02:11.57	-0.46
1500	03:29.46	03:29.16	-0.14	03:30.23	+0.37	03:31.63	+1.03	03:29.09	-0.18
1609	03:46.32	03:47.33	+0.45	03:47.18	+0.38	03:49.18	+1.26	03:46.62	+0.13
2000	04:50.81	04:52.50	+0.58	04:48.07	-0.94	04:52.34	+0.53	04:51.15	+0.12
3000	07:32.10	07:39.16	+1.56	07:25.84	-1.38	07:34.92	+0.62	07:32.41	+0.07
5000	12:58.39	13:12.49	+1.81	13:04.46	+0.78	13:03.24	+0.82	12:59.38	+0.13
10000	27:13.81	27:05.82	-0.49	27:32.87	+1.17	27:00.38	-0.82	27:13.51	-0.02
21100	1:00:55.00	57:55.81	-4.90	1:00:49.44	-0.15	59:29.22	-2.35	1:00:35.14	-0.54
42195	2:07:12.00	1:56:31.62	-8.39	2:06:33.66	-0.50	2:08:44.79	+1.22	2:07:39.60	+0.36

distance	time	CP model	error [%]	PT model	error [%]	dIP model	error [%]	MIT model	error [%]
800	01:40.91	01:33.77	-7.07	01:41.32	+0.41	01:40.20	-0.70	01:41.71	+0.79
1000	02:11.96	02:05.89	-4.60	02:11.49	-0.36	02:10.77	-0.90	02:10.58	-1.04
1500	03:26.00	03:26.18	+0.09	03:27.05	+0.51	03:28.12	+1.03	03:26.09	+0.04
1609	03:43.13	03:43.68	+0.25	03:43.53	+0.18	03:45.08	+0.87	03:43.08	-0.02
2000	04:44.79	04:46.47	+0.59	04:42.74	-0.72	04:46.09	+0.46	04:45.40	+0.21
3000	07:20.67	07:27.05	+1.45	07:15.31	-1.22	07:22.99	+0.53	07:20.91	+0.05
5000	12:37.35	12:48.20	+1.43	12:41.29	+0.52	12:39.51	+0.29	12:36.66	-0.09
10000	26:17.53	26:11.10	-0.41	26:33.79	+1.03	26:05.02	-0.79	26:14.99	-0.03
21100	58:01.00	55:53.52	-3.66	58:17.54	+0.48	57:11.12	-1.43	58:05.83	+0.14
42195	2:01:39.00	1:52:20.93	-7.65	2:00:36.40	-0.86	2:02:35.30	+0.77	2:01:34.01	-0.07

FIG. 3 Application of four mathematical models to men running world records from 1987 and 2020: Log-normal plot of the 'running curves' predicted by the models (average velocity \bar{v} as function of race distance d , in units of v_m and $d_c = v_m t_c$ given in the plot legend) and actual race data (red dots). The tables summarize the actual and predicted race times, along with the relative errors in percent.

distance	time	CP model	error [%]	PT model	error [%]	dIP model	error [%]	MIT model	error [%]
1500	03:39.25	03:31.80	-3.40	03:40.26	+0.46	03:37.58	-0.76	03:39.25	+0.00
3000	07:41.33	07:41.66	+0.07	07:34.21	-1.54	07:43.99	+0.58	07:38.67	-0.58
5000	13:02.86	13:14.80	+1.53	13:09.54	+0.85	13:16.00	+1.68	13:08.15	+0.68
10000	27:12.47	27:07.65	-0.30	27:33.80	+1.31	27:20.58	+0.50	27:25.68	+0.81
21097	1:01:45.00	57:56.09	-6.18	1:00:51.38	-1.45	59:54.66	-2.98	1:00:45.21	-1.61
42195	2:06:36.00	1:56:30.40	-7.97	2:07:01.20	+0.33	2:08:12.86	+1.28	2:07:26.14	+0.66

distance	time	CP model	error [%]	PT model	error [%]	dIP model	error [%]	MIT model	error [%]
1500	03:33.20	03:23.46	-4.57	03:33.94	+0.34	03:31.20	-0.94	03:33.10	-0.05
3000	07:27.66	07:30.51	+0.64	07:26.12	-0.34	07:32.61	+1.11	07:29.73	+0.46
3218.68	08:07.39	08:06.52	-0.18	08:01.92	-1.12	08:07.99	+0.12	08:05.23	-0.44
5000	12:46.53	12:59.91	+1.75	12:55.80	+1.21	12:57.53	+1.44	12:51.56	+0.66
10000	26:49.02	26:43.41	-0.35	26:57.17	+0.51	26:41.56	-0.46	26:42.72	-0.39
21097	59:25.00	57:11.09	-3.76	58:58.62	-0.74	58:12.28	-2.04	58:48.35	-1.03
42195	2:01:39.00	1:55:05.94	-5.39	2:01:48.73	+0.13	2:02:48.65	+0.95	2:02:34.67	+0.76

FIG. 5 (a) Application of four mathematical models to personal records of elite marathon runners (Antonio Pinto, Eliud Kipchoge): Log-normal plot of the 'running curves' predicted by the models (average velocity \bar{v} as function of race distance d , in units of v_m and $d_c = v_m t_c$ given in the plot legend) and actual race data (red dots). The tables summarize the actual and predicted race times, along with the relative errors in percent.

distance	time	CP model	error [%]	PT model	error [%]	dIP model	error [%]	MIT model	error [%]
1500	03:40.14	03:30.81	-4.24	03:40.93	+0.36	03:37.48	-1.21	03:40.14	+0.00
3000	07:40.67	07:42.85	+0.47	07:36.52	-0.90	07:47.68	+1.52	07:40.67	-0.00
5000	13:16.42	13:18.90	+0.31	13:13.50	-0.37	13:24.72	+1.04	13:12.92	-0.44
10000	27:04.54	27:19.03	+0.89	27:40.64	+2.22	27:40.89	+2.24	27:34.04	+1.82
15000	41:29.00	41:19.17	-0.40	42:31.69	+2.52	42:16.77	+1.92	42:25.21	+2.26
16093.4	46:41.00	44:22.89	-4.93	45:48.98	-1.86	45:31.07	-2.50	45:43.06	-2.07
20000	58:20.00	55:19.30	-5.16	57:39.85	-1.15	57:13.80	-1.89	57:37.14	-1.22
21097	1:02:05.00	58:23.62	-5.94	1:01:01.00	-1.72	1:00:33.61	-2.45	1:00:59.48	-1.76
42195	2:06:14.00	1:57:28.64	-6.94	2:07:07.95	+0.71	2:08:43.21	+1.97	2:07:46.93	+1.23

distance	time	CP model	error [%]	PT model	error [%]	dIP model	error [%]	MIT model	error [%]
800	01:49.35	01:37.69	-10.66	01:49.37	+0.02	01:47.02	-2.13	01:49.20	-0.14
1500	03:31.76	03:30.09	-0.79	03:34.56	+1.32	03:35.83	+1.92	03:33.07	+0.62
2000	04:52.86	04:50.38	-0.85	04:49.56	-1.13	04:54.19	+0.45	04:49.49	-1.15
3000	07:25.09	07:30.95	+1.32	07:21.03	-0.91	07:31.83	+1.51	07:25.09	-0.00
3218.68	08:01.08	08:06.07	+1.04	07:56.24	-1.01	08:06.44	+1.12	07:59.55	-0.32
5000	12:39.36	12:52.10	+1.68	12:46.64	+0.96	12:50.23	+1.43	12:45.27	+0.78
10000	26:22.75	26:14.96	-0.49	26:46.51	+1.50	26:24.07	+0.08	26:39.32	+1.05
16093.4	44:24.00	42:33.39	-4.15	44:23.01	-0.04	43:33.24	-1.91	44:15.99	-0.30
20000	55:48.00	53:00.68	-5.00	55:54.06	+0.18	54:57.24	-1.52	55:49.62	+0.05
21097	58:55.00	55:56.83	-5.04	59:09.71	+0.42	58:13.03	-1.19	59:06.27	+0.32
25000	1:11:37.00	1:06:23.55	-7.29	1:10:50.90	-1.07	1:10:03.89	-2.17	1:10:51.77	-1.05
42195	2:03:59.00	1:52:24.59	-9.33	2:03:35.91	-0.31	2:07:49.12	+3.09	2:04:06.73	+0.10

FIG. 5 (b) Application of four mathematical models to personal records of elite marathon runners (Felix Limo, Haile Gebrselassie): Log-normal plot of the 'running curves' predicted by the models (average velocity \bar{v} as function of race distance d , in units of v_m and $d_c = v_m t_c$ given in the plot legend) and actual race data (red dots). The tables summarize the actual and predicted race times, along with the relative errors in percent.

distance	time	CP model	error [%]	PT model	error [%]	dIP model	error [%]	MIT model	error [%]
800	01:48.24	01:35.74	-11.55	01:47.43	-0.75	01:46.05	-2.02	01:47.52	-0.67
1500	03:28.81	03:30.22	+0.67	03:33.14	+2.08	03:35.60	+3.25	03:31.18	+1.14
3218.68	08:03.40	08:11.29	+1.63	07:56.54	-1.42	08:08.16	+0.98	08:01.04	-0.49
5000	12:53.11	13:02.60	+1.23	12:49.49	-0.47	12:53.77	+0.09	12:51.26	-0.24
10000	26:46.57	26:40.29	-0.39	26:57.13	+0.66	26:31.30	-0.95	26:51.68	+0.32
21097	59:32.00	56:55.07	-4.39	59:39.84	+0.22	58:18.11	-2.07	59:33.22	+0.03
42195	2:05:11.00	1:54:25.40	-8.60	2:04:48.67	-0.30	2:06:34.96	+1.12	2:05:02.35	-0.12

distance	time	CP model	error [%]	PT model	error [%]	dIP model	error [%]	MIT model	error [%]
800	01:47.43	01:39.51	-7.37	01:47.79	+0.33	01:47.18	-0.24	01:47.43	+0.00
3000	07:49.80	07:51.18	+0.29	07:43.98	-1.24	07:50.84	+0.22	07:48.88	-0.20
3218.68	08:26.71	08:28.13	+0.28	08:21.43	-1.04	08:27.34	+0.12	08:24.88	-0.36
5000	13:18.60	13:29.07	+1.31	13:28.36	+1.22	13:26.01	+0.93	13:22.44	+0.48
10000	27:39.14	27:33.79	-0.32	28:04.09	+1.50	27:37.21	-0.12	27:45.63	+0.39
21097	1:01:14.00	58:48.55	-3.96	1:01:15.79	+0.05	1:00:18.34	-1.52	1:01:03.67	-0.28
42195	2:07:13.00	1:58:12.92	-7.08	2:06:07.20	-0.86	2:08:03.76	+0.67	2:07:10.00	-0.04

FIG. 5 (c) Application of four mathematical models to personal records of elite marathon runners (Mo Farah, Steve Jones): Log-normal plot of the 'running curves' predicted by the models (average velocity \bar{v} as function of race distance d , in units of v_m and $d_c = v_m t_c$ given in the plot legend) and actual race data (red dots). The tables summarize the actual and predicted race times, along with the relative errors in percent.

Reviewers' Comments:

Reviewer #1:

Remarks to the Author:

Given that my involvement to the review process started at a later stage, I will avoid providing detailed comments on each section as I would typically do. However, I have read the manuscript in detail. Although I appreciate the value of exploring big data sets, I have a major concern with this manuscript as I do not think that any link can be made to physiological responses to exercise, when no physiological measures have been extracted. Additionally, I would like to mention that this manuscript is quite difficult to read, and that the authors should make an effort to improve the flow and logical order of the presentation. Regardless, please find below some general comments that I would hope will help the authors reflecting further on this manuscript.

I think that the authors are not fully aware of the type of testing that takes place in many laboratories. I understand that they need to highlight the relevance of "real-world" data, and that laboratory settings have limitations. However, there are many experimental studies that have produced very solid performance data that, even though they do not belong to the "real world" category, they offer information that the "real world" conditions will never provide. I fully agree that the best measure of performance is performance itself. From a performance perspective, I do not care about who has the greatest VO₂max or critical intensity of exercise. I care about who runs faster. Then, from a mechanistic perspective, I bring people to the lab to try to understand why differences in performance exist, but not necessarily to make people faster. The authors stated "The undeniable fact that the best test of running performance is an actual race and not laboratory tests" is only partly true. It is the best test to measure performance. However, it is not the best test to evaluate physiological responses and to elucidate the mechanisms that control the final performance. I think that the point that I am trying to make is that, at least to a given extent, the authors seem to be misrepresenting what happens in a laboratory setting.

From what I have read in this manuscript, there is nothing that connects its content to physiological responses to exercise (which are often mentioned in this document). I could accept the claim that this analysis can help establishing non-physiological outcomes that could potentially help improving performance. However, there is no physiological value that can be seriously considered in this data set. At least in my view, the model requires accepting assumptions that might make some sense, but that are not necessarily correct. The authors seem to have almost a dislike for physiological evaluations. I am fine with that. However, there is no point in discussing physiology when no physiological outcomes are presented. I do not feel comfortable with all the assumptions that need to be accepted to believe some of the key components of the analysis (e.g., MAP).

Once again, the authors might have gotten it right in terms of some predictors of performance. The problem is that we will never know as no real physiological data were collected. Perhaps, performing some physiological testing in a sub-sample of participants would add validity to the project. However, the authors have already disregarded this possibility when responding to other reviewers. In relation to this, I was interested in some responses. I am presenting below just a few examples:

- The authors indicated that "As far as heart rate is concerned, our data set does not contain heart rate data for all runs and athletes as not all runners who wore a GPS watch wear a heart rate monitor (chest strap). But even if this data would be available, there remains an important unknown: the maximal heart rate of the athlete which varies substantially among individuals and cannot be determined accurately and easily from age-based formulas. Without the maximal heart rate the important relative effort (the quantity p in our model) cannot be determined accurately." I would accept that the age-based formulas are not ideal, but they can be a good approximation. Additionally, the authors have plenty of data from the participants and I am sure that there has to be some high intensity interval or sprint training, or high intensity constant speed session from

which HRmax could be derived. I mean, I would be the first arguing that, even if you had the actual HRmax, there are clear limitations with this approach. However, what I find a bit surprising is that the authors are willing to accept a lot of assumptions for other parameters in their model, but then they are too concerned about not getting the HRmax 100% right. This is surprising to me.

- The authors argued that "Maximal tests in laboratory are difficult to repeat, possibly due to lack of motivation to go all-out without opponent or competition or even prize money to win. As a result, the coefficient of variation may be as high as 25%". Let's clarify that performance outcomes have large variability in both the lab and on the field, but that the variability is greatly reduced with longer durations of performance. Additionally, if the lack of motivation because of the prize money is an issue, then the author should eliminate most of these data because the vast majority of the performances in the people that the authors evaluated are not worth any money. Most people are engaged for other reasons and most of them would perform as well in the lab as they do in the "real world". I am not convinced by this line of argumentation.

- Then the authors stated that "Running mechanics varies considerably between treadmill and over ground running...One reason may difficulty to simulate wind resistance". In fact, there are portable devices to test people in the "real world". I know, the conditions will be slightly different. However, nothing is perfect (and this includes the assumptions in the model that is presented by the authors).

- Finally, the authors said, "Maximal laboratory tests are short-lasting and therefore fail to account for reduction in running economy and subsequent increase in oxygen consumption at given speed that occurs over long-distance running." Why would this need to be the case? I just read a paper in which participants performed quite long incremental tests achieving the same VO₂max as in the shorter tests (J Appl Physiol 2019; 127(6):1519-1527). Maximal tests do not need to be short. Testing protocols are adapted to what one wants to evaluate. This type of comments makes me feel that the authors might not be very familiar with laboratory testing.

As a side comment, I would say that the speed and endurance relationship presented in this document are quite similar to what is typically measured in the lab. So why emphasizing so much the idea that field data are better than lab data? Also, the fact that from training data one can predict performance is pretty obvious. What one can do in a race reflects what one can do in training. I know it is nice to confirm this with data, but there is nothing novel in this finding.

As a final comment, I would like to say that I do not think that the authors have a full appreciation of the relevance that exercise intensity domains and their corresponding boundaries (i.e., thresholds) have in performance. I understand that measurements of VO₂ and exercise thresholds have been largely bastardized in the world of exercise testing (to which the authors contribute by arbitrarily assigning names to parameters such as MAP or LT without having any physiological way of justifying them in this study). However, when things are done properly, very precise quantification of the metabolic stress of the system can be made. Unlike what the authors insinuate, these evaluations consider economy, fatigue, substrate depletion, etc. to make predictions about performance. All I am trying to say is that the authors might have an interesting story in relation to non-physiological predictors of running performance. However, they should be very careful with not overreaching beyond of what their data can say.

Reviewer #2:

Remarks to the Author:

I appreciate the authors' detailed rebuttal and the appendix that they have included to compare their model to other similar models. While the paper is improved, it is still hard to follow and ascertain exactly what the novel insights are. I think that many exciting findings have resulted from the analysis, but as the paper is presently written, many of the key insights do not stand out to the reader.

It is also not clear whether the focus of the paper is to provide additional evidence to validate their previously published model or to show some of the novel insights that applying their model to the dataset can generate. It might be possible to do both things, but this should be framed more explicitly at the beginning and then discussed more explicitly in the results. If the goal is to provide additional support for their previous model, then the comparisons that they include in the appendix of the rebuttal would at least be helpful to include as supplementary material. I am personally more interested in a focus on the insights gained from the application of their model to the real-world dataset. If this is the desired focus, this should be made more clear in the manuscript. Even in this case, the comparisons to other models would still provide confidence that the author's model is reasonable and thus could still be helpful to include in supplementary material.

These and other comments are discussed in more detail below.

A couple general comments on the review process:

Line numbers are very helpful in the manuscript review process; then the authors can note line numbers where changes have been made in the response to reviewers. As a reviewer, I can also provide specific locations relevant for my comments. An annotated version of the manuscript showing exactly where changes have been made (e.g., via highlighting) is also very helpful to me as a reviewer.

Title

The paper should include a more meaningful title that highlights the specific novelty of the present work. The terms "novel" and "insights" do not convey much information about the present work. The term "novel" should be removed at minimum, as I believe is policy at least for Nature. The authors were also not performing data mining by most definitions of the term, since they were using a pre-existing physiology-based model (as a side note, I think this approach is preferable, in general, to a naive data mining one). Instead, the real-world or free-living nature of the data is relevant to highlight in the title. The size of the data is also worth noting, as the title already does.

Abstract

(1) "We derived two variables that explain race performance: maximal aerobic power and endurance capability. Inclusion of endurance, which describes the decline in sustainable power over duration, offers novel insights to performance analysis since a realistic estimate of this parameter is impossible in conventional laboratory testing."

The mathematical model that the authors use was presented in the authors' previously published paper. The abstract gives the impression that the mathematical model is something newly-created for the present paper. Please revise to make the novelty of the current paper more clear (i.e., the application of the model to free-living data and interpretation of the extracted parameters).

(2) The abstract is much more clear than in the previous version, but it still does not include specific results. Novel insights are mentioned. But what were these novel insights?

Introduction

(3) In general, the introduction (along with other parts of the paper) is unnecessarily negative about in-lab testing. Both in-lab and out-of-lab testing have strengths and weaknesses and these could be acknowledged in a more even-handed way.

(4) "important insights for a variety of populations ranging from elite athletes over recreational exercisers to patients in rehabilitation"

change over -> to

(5) "These approach predict that the average racing velocity tends to a constant value with

increasing race distance which contradicts observation”

Approach -> approaches

Tends to an -> tends to be a

(6) “Several empirical and physiological models have been put forward for explaining running world records in terms of a few physiological parameters.”

Start a new paragraph here.

(7) “Our minimal and universal model characterizes a runner’s physiology by two parameters that measure endurance capability and the velocity requiring maximal aerobic power output”

The authors should make more clear that the model has already been proposed and evaluated with some data from (real-world) races. The application of the model to the present dataset (and to training data?) is what makes the current paper new. The previous paper by the authors should be mentioned and cited in the introduction, for example. This should also be made more clear in the last paragraph of the introduction that lays out the goals for the paper.

Results

(8) “Universal Performance Model” section: The authors should more directly state that they are using the model that they present in a previous publication. Something like:

(1) In previous work we developed a model that does X. To summarize, this model (describe the key features of the model). For more details, see XXXX.

(2) Here we do XXXX with the model.

If there are differences between the author’s model published previously and the one in the present model, please make these differences more clear.

(9) The results section and paper in general would also benefit from a tighter focus on the key, novel findings of the paper. For example, below are some excerpts from the paper that are novel, but don’t stand out in the present draft. Focusing paragraphs in the results on each of these topics, would be helpful. Specific paragraphs could be focused around asking the associated questions and discussing the study results. The key findings could also be explicitly enumerated in the discussion.

- For all RS with three and more races (N=12,309), the mean error between model prediction and actual race time was only 2.0% ... As a function of physiological parameters, in the most likely parameter range the model predicted the marathon performance with an overall accuracy of better than 10%.

- The “one-hour utilization” ratio $p1hU = v1hU / v_m$ had been estimated previously from laboratory measurements and races for a smaller group of 18 male LDR to be approximately 0.82 ± 0.05 . Strikingly, our findings from the running data for ~ 14,000 subjects corroborate this range without any invasive measurements, as demonstrated in Fig. 2(c).

- Our findings demonstrate the strong sensitivity of performance to endurance. For example, a runner with a velocity of $v_m = 5\text{m/sec}$ can improve their marathon time from 3h27min38sec to 2h53min8sec by doubling endurance from $EI = 3$ to $EI = 6$ (corresponding to a change in the “one-hour utilization” from 79% to 87% of VO_{2max}), without any change in VO_{2max} or RE.

- We observed an initial linear increase of EI with TRIMP, a plateau around $EI = 7.5 \pm 2$ for TRIMP ~ 25,000, and a statistically significant final drop which may be due to over-training. This result suggests that there is an optimal TRIMP per TS, and the corresponding maximal endurance enables a close to optimal marathon race time for a given velocity v_m (see Fig. 3(a)).

(10) Minimize the use of acronyms where possible in the text to make it easier for readers to understand the paper. I suggest you remove the following:

- RS (racing season)
- TS (training season)
- RE (running economy)
- LDR (long distance runners?)

If the abbreviations are needed in a figure/table they are OK to use there, as long as they are defined in the caption.

(11) "by matching them with an universal, i.e., subject independent model"

An universal -> a universal

A comma is needed after "model"

(12) "Our minimal model introduces effective parameters by measuring" It is not clear what the authors mean by "effective".

(13) "observations made by Hill in running world records"

Reword to make it clear that it wasn't Hill who was running the world records :-).

(14) "Fig. 3 first shows a color coded plot of Tmarathon as function of the physiological parameters."

This type of sentence is a better fit for a caption. In the Results it is preferable to describe specific findings. There are several instances of this in the Results.

(15) "To investigate the predictive power of our model in more detail, we applied our model also the RS with the marathon performance excluded"

A word is missing from this sentence.

(16) "Consistent and inconsistent runners can be identified from the relative difference between our model estimates and actual race times." A better topic sentence (that covers the main focus of the paragraph) is needed to improve the logical flow of this section of the results. In general, a careful review of the entire paper to ensure each paragraph has a clear topic sentence would improve the quality of the manuscript.

Discussion

(17) First paragraph: this should be broken into multiple paragraphs. The discussion of the limitations would be a natural split point.

(18) "This is an important advance over physiological testing in the laboratory where the required maximal effort is impossible to motivate for a distance of 20km or longer."

I don't think the authors intend to mean that there is no use for lab-based testing. This is another place where the authors could soften their language. (e.g., important advance -> important complement).

In general, the primary point that stands out from the discussion is that the real-world data is a big improvement over lab testing. I don't think this is the most important point (as lab-based testing in a controlled environment still has great value). I would instead focus more on reviewing the specific new insights about running, training, and performance that were gleaned from the analysis.

Methods

(19) "Only TS with 30 or more runs were considered."

What is the rationale for this choice? Was there any requirement from the minimum chronological

length of the training season? Was there any sensitivity to these or other threshold choices discussed in the paragraph?

(20) Check for redundancy between material included in the Methods and Results

(21) The following passage is a better fit for the results or discussion than the Methods.

For our two parameter model, the quality of the fitting could be probed for all RS with more than two races. For those RS we found a rather low average error of only 2.0% between the computed and actual race times. Another applicability test of our model is the estimation of the marathon finishing time from equation(1) when the parameters v_m and l are obtained from the RS without the marathon. Given all the possible uncertainties in marathon racing that are beyond the control of this study (e.g. weather, course profile, motivation of the athlete), the predictive power reflected by the results for marathon finishing time estimate in Fig. 4 is rather satisfying

Reviewer #3:

Remarks to the Author:

The authors have provided detailed and convincing responses to most of the questions and comments that I forwarded to them. In particular, I am convinced by their response on the relationship between their model and the critical power model. However, this did not translate into a modification of the article accounting entirely for their responses to the reviewer. This is a pity. The changes in the manuscript are minor and clearly inadequate. I would like to see the reasoning that the author developed in replying to the reviewer's comments more adequately integrated in the manuscript, especially in the discussion, and I hope the authors will show more consideration for the suggested references and comments. To respond is good, but it is not enough.

Rebuttal letter

“Human running performance from real-world big data” (NCOMMS-20-02292)

Please find below our point-to-point answers to the reviewer comments (C: comment, A: answer). All changes in the manuscript are marked by colour highlighting (deleted text in red, newly added text in blue). Also, we have included line numbers in the manuscript (colour coded version) in order to make reference to changes in the point-by-point rebuttal letter.

Answer to Reviewer #1

We thank the reviewer for her/his time spent looking over our manuscript and their comments that we address point-by-point in the following.

C *I think that the authors are not fully aware of the type of testing that takes place in many laboratories. I understand that they need to highlight the relevance of real-world data, and that laboratory settings have limitations. However, there are many experimental studies that have produced very solid performance data that, even though they do not belong to the real world category, they offer information that the real world conditions will never provide. I fully agree that the best measure of performance is performance itself. From a performance perspective, I do not care about who has the greatest VO₂max or critical intensity of exercise. I care about who runs faster. Then, from a mechanistic perspective, I bring people to the lab to try to understand why differences in performance exist, but not necessarily to make people faster. The authors stated The undeniable fact that the best test of running performance is an actual race and not laboratory tests is only partly true. It is the best test to measure performance. However, it is not the best test to evaluate physiological responses and to elucidate the mechanisms that control the final performance. I think that the point that I am trying to make is that, at least to a given extent, the authors seem to be misrepresenting what happens in a laboratory setting.*

A We agree that our presentation was not balanced between laboratory testing and our approach. This is regrettable because that is not how we think. Therefore, we now highlight how wearables can **complement** laboratory testing by expanding the size of population that can be tested. We also compare strengths and weaknesses of both approaches. Please see lines 46ff, 61ff, 102ff, 345ff.

C *From what I have read in this manuscript, there is nothing that connects its content to physiological responses to exercise (which are often mentioned in this document). I could accept the claim that this analysis can help establishing non-physiological outcomes that could potentially help improving performance. However, there is no physiological value that can be seriously considered in this data set. At least in my view, the model requires accepting assumptions that might make some sense, but that are not necessarily correct. The authors seem to have almost a dislike for physiological evaluations. I am fine with that. However, there is no point in discussing physiology when no physiological outcomes are presented. I do not feel comfortable with all the assumptions that need to be accepted to believe some of the key components of the analysis (e.g., MAP).*

A We have replaced throughout the manuscript “physiological parameters” by indices of performance (aerobic power index and endurance index), extracted from running exercise data using our model. We would like to point out that our model makes assumptions that are also contained in other models proposed by exercise physiologists (e.g. Monod & Scherrer, di Prampero, Peronnet & Thibault, see our detailed comparison in the appendix to our last rebuttal letter). Hence, we believe that the key parameters of our model do have some physiological meaning. However, in order to avoid any confusion and to not make unnecessary assumptions, we now refer to our parameters as “performance indices” and just state to which physiological variables they might be related. Please see lines 96ff.

- C *Once again, the authors might have gotten it right in terms of some predictors of performance. The problem is that we will never know as no real physiological data were collected. Perhaps, performing some physiological testing in a sub-sample of participants would add validity to the project. However, the authors have already disregarded this possibility when responding to other reviewers.*
- A We agree that physiological testing on a smaller sample of subjects would be very useful. We plan to carry out such testing for a new group of subject in the future, to compare our model parameters to actual lab measurements. It should be noted that also previous models (e.g. Peronnet & Thibault) have been applied only to world running records to extract physiological parameters without a direct comparison to lab tests for these athletes.
- C *The authors indicated that As far as heart rate is concerned, our data set does not contain heart rate data for all runs and athletes as not all runners who wore a GPS watch wear a heart rate monitor (chest strap). But even if this data would be available, there remains an important unknown: the maximal heart rate of the athlete which varies substantially among individuals and cannot be determined accurately and easily from age-based formulas. Without the maximal heart rate the important relative effort (the quantity p in our model) cannot be determined accurately. I would accept that the age-based formulas are not ideal, but they can be a good approximation. Additionally, the authors have plenty of data from the participants and I am sure that there has to be some high intensity interval or sprint training, or high intensity constant speed session from which HR_{max} could be derived. I mean, I would be the first arguing that, even if you had the actual HR_{max} , there are clear limitations with this approach. However, what I find a bit surprising is that the authors are willing to accept a lot of assumptions for other parameters in their model, but then they are too concerned about not getting the HR_{max} 100% right. This is surprising to me.*
- A Our point is that HR would not add any *additional* benefit to the extraction of our model parameters. When maximal and resting HR for each runner are known, the entire analysis could be based on HR instead of running velocity, yielding an expression for the maximal duration over which a given HR could be sustained, $T_{max}(HR)$. Hence the parameters v_m and E_l could be determined from observed relations between velocity and HR, and the average HR sustained during maximal effort of a given duration (races). However, an unpublished study that we performed previously on a much smaller number of subjects (20) showed that HR fluctuates more strongly than velocity, presumably due to weather conditions, non-running related stress, nutrition status, sleep status, etc. In addition, there is always a time delay between a rise (or fall) in velocity and HR which requires the exclusion of time windows with this hysteresis effects. The accuracy of our model for race time predictions was on average 2%. This is definitely better than the typical error for age-based formulas for maximal HR. All these considerations led us to use velocity instead of HR in our data analysis.
- C *The authors argued that Maximal tests in laboratory are difficult to repeat, possibly due to lack of motivation to go all-out without opponent or competition or even prize money to win. As a result, the coefficient of variation may be as high as 25%. Lets clarify that performance outcomes have large variability in both the lab and on the field, but that the variability is greatly reduced with longer durations of performance. Additionally, if the lack of motivation because of the prize money is an issue, then the author should eliminate most of these data because the vast majority of the performances in the people that the authors evaluated are not worth any money. Most people are engaged for other reasons and most of them would perform as well in the lab as they do in the real world. I am not convinced by this line of argumentation.*
- A In the revised version of the manuscript we do not state that poor repeatability would compromise laboratory test results. We would like to point out that not only prize money is motivation but also competing against friends, team members, or for something like age group win etc., i.e., real-world situations.

- C *Then the authors stated that Running mechanics varies considerably between treadmill and over ground running...One reason may difficulty to simulate wind resistance. In fact, there are portable devices to test people in the real world. I know, the conditions will be slightly different. However, nothing is perfect (and this includes the assumptions in the model that is presented by the authors).*
- A Our data comes from consumer-product based measurements (GPS watches) and more advanced portable devices were not available to the huge group of runners monitored. We agree on the general possibility of more advanced measurements in the "real world". However, these additional data are not relevant to our model as an input, and they would impose limitations on the number of available subjects.
- C *Finally, the authors said, Maximal laboratory tests are short-lasting and therefore fail to account for reduction in running economy and subsequent increase in oxygen consumption at given speed that occurs over long-distance running. Why would this need to be the case? I just read a paper in which participants performed quite long incremental tests achieving the same VO₂max as in the shorter tests (J Appl Physiol 2019; 127(6):1519-1527). Maximal tests do not need to be short. Testing protocols are adapted to what one wants to evaluate. This type of comments makes me feel that the authors might not be very familiar with laboratory testing.*
- A We are not saying that VO₂max is declining with duration. We are only saying that running economy (energy cost of running) deteriorates with duration. This is based on results in Ref. 18, 19 and 20. However, in order to provide in this context a better balance between lab testing and our approach, we have modified the paragraph with this statement, see lines 46 – 68.
- C *As a side comment, I would say that the speed and endurance relationship presented in this document are quite similar to what is typically measured in the lab. So why emphasizing so much the idea that field data are better than lab data? Also, the fact that from training data one can predict performance is pretty obvious. What one can do in a race reflects what one can do in training. I know it is nice to confirm this with data, but there is nothing novel in this finding.*
- A We are glad to hear that what is measured typically in the lab is quite similar to our model findings. We believe that the novel part of findings are *quantitative* relations between our model indexes and training volume and intensity for a very large group of runners, yielding also good statistics for typical variations in these relations. We do not think that field data are better than lab testing. We have made clear in the revised manuscript that our approach is **complementary** to lab testing, see lines 324 – 326.
- C *As a final comment, I would like to say that I do not think that the authors have a full appreciation of the relevance that exercise intensity domains and their corresponding boundaries (i.e., thresholds) have in performance. I understand that measurements of VO₂ and exercise thresholds have been largely bastardized in the world of exercise testing (to which the authors contribute by arbitrarily assigning names to parameters such as MAP or LT without having any physiological way of justifying them in this study). However, when things are done properly, very precise quantification of the metabolic stress of the system can be made. Unlike what the authors insinuate, these evaluations consider economy, fatigue, substrate depletion, etc. to make predictions about performance. All I am trying to say is that the authors might have an interesting story in relation to non-physiological predictors of running performance. However, they should be very careful with not overreaching beyond of what their data can say.*
- A We accept this criticism. We have now renamed model parameters to aerobic power index and endurance index to differentiate them from laboratory parameters. Overall, we have rewritten our manuscript in order to provide a more balanced presentation of what our model predicts and the concepts and measurements in the world of exercise testing in the lab.

Rebuttal letter

“Human running performance from real-world big data” (NCOMMS-20-02292)

Please find below our point-to-point answers to the reviewer comments (C: comment, A: answer). All changes in the manuscript are marked by colour highlighting (deleted text in red, newly added text in blue). Also, we have included line numbers in the manuscript (colour coded version) in order to make reference to changes in the point-by-point rebuttal letter.

Answer to Reviewer #2

We thank the reviewer for her/his time spent again looking over our manuscript and their very detailed comments and suggestions that we address point-by-point in the following.

C *It is also not clear whether the focus of the paper is to provide additional evidence to validate their previously published model or to show some of the novel insights that applying their model to the dataset can generate. It might be possible to do both things, but this should be framed more explicitly at the beginning and then discussed more explicitly in the results. If the goal is to provide additional support for their previous model, then the comparisons that they include in the appendix of the rebuttal would at least be helpful to include as supplementary material. I am personally more interested in a focus on the insights gained from the application of their model to the real-world dataset. If this is the desired focus, this should be made more clear in the manuscript. Even in this case, the comparisons to other models would still provide confidence that the authors model is reasonable and thus could still be helpful to include in supplementary material.*

A The main aim of our work is “to show some of the novel insights that applying their model to the dataset can generate”. We have made this more clear in the revised manuscript, please see lines 96ff. The model comparison of our previous rebuttal letter is not included in this work as it would go much beyond its scope. Instead, we are in the process of completing a separate publication on a detailed comparison of mathematical models for running performance which shall include the results shown in our rebuttal letter.

C *Title: The paper should include a more meaningful title that highlights the specific novelty of the present work. The terms novel and insights do not convey much information about the present work. The term novel should be removed at minimum, as I believe is policy at least for Nature. The authors were also not performing data mining by most definitions of the term, since they were using a pre-existing physiology-based model (as a side note, I think this approach is preferable, in general, to a naive data mining one). Instead, the real-world or free-living nature of the data is relevant to highlight in the title. The size of the data is also worth noting, as the title already does.*

A We have modified the title accordingly.

C *(1) We derived two variables that explain race performance: maximal aerobic power and endurance capability. Inclusion of endurance, which describes the decline in sustainable power over duration, offers novel insights to performance analysis since a realistic estimate of this parameter is impossible in conventional laboratory testing. The mathematical model that the authors use was presented in the authors previously published paper. The abstract gives the impression that the mathematical model is something newly-created for the present paper. Please revise to make the novelty of the current paper more clear (i.e., the application of the model to free-living data and interpretation of the extracted parameters).*

A We have revised the abstract accordingly.

C (2) *The abstract is much more clear than in the previous version, but it still does not include specific results. Novel insights are mentioned. But what were these novel insights?*

A We now described the novel insights more specifically.

C (3) *In general, the introduction (along with other parts of the paper) is unnecessarily negative about in-lab testing. Both in-lab and out-of-lab testing have strengths and weaknesses and these could be acknowledged in a more even-handed way.*

A We have given a more balanced presentation of in-lab and out-of-lab testing, please see lines 46ff, 61ff, 102ff, 345ff.

C (4) *important insights for a variety of populations ranging from elite athletes over recreational exercisers to patients in rehabilitation: change over → to*

A done.

C (5) *These approach predict that the average racing velocity tends to an constant value with increasing race distance which contradicts observation: Approach → approaches, Tends to an → tends to be a*

A done.

C (6) *Several empirical and physiological models have been put forward for explaining running world records in terms of a few physiological parameters.: Start a new paragraph here.*

A done.

C (7) *Our minimal and universal model characterizes a runners physiology by two parameters that measure endurance capability and the velocity requiring maximal aerobic power output. The authors should make more clear that the model has already been proposed and evaluated with some data from (real-world) races. The application of the model to the present dataset (and to training data?) is what makes the current paper new. The previous paper by the authors should be mentioned and cited in the introduction, for example. This should also be made more clear in the last paragraph of the introduction that lays out the goals for the paper.*

A We have modified the introduction accordingly, and referenced our previous paper. Please see lines 77 – 83.

C (8) *Universal Performance Model section: The authors should more directly state that they are using the model that they present in a previous publication. Something like: (1) In previous work we developed a model that does X. To summarize, this model . (describe the key features of the model). For more details, see XXXX. (2) Here we do XXXX with the model. If there are differences between the authors model published previously and the one in the present model, please make these differences more clear.*

A We modified this section accordingly (lines 108 – 124). There is no difference with the model itself published previously. One of the parameters of the model (t_c) was fixed at 6 minutes, as we had explained already in the previous version of the manuscript.

C (9) *The results section and paper in general would also benefit from a tighter focus on the key, novel findings of the paper. For example, below are some excerpts from the paper that are novel, but dont stand out in the present draft. Focusing paragraphs in the results on each of these topics, would be helpful. Specific paragraphs could be focused around asking the associated questions and discussing the study results. The key findings could also be explicitly enumerated in the discussion.*

- For all RS with three and more races ($N=12,309$), the mean error between model prediction and actual race time was only 2.0%. As a function of physiological parameters, in the most likely parameter range the model predicted the marathon performance with an overall accuracy of better than 10%.
- The one-hour utilization ratio $p_{1hU} = v_{1hU}/v_m$ had been estimated previously from laboratory measurements and races for a smaller group of 18 male LDR to be approximately 0.82 ± 0.05 . Strikingly, our findings from the running data for $\sim 14,000$ subjects corroborate this range without any invasive measurements, as demonstrated in Fig. 2(c).
- Our findings demonstrate the strong sensitivity of performance to endurance. For example, a runner with a velocity of $v_m = 5\text{m/sec}$ can improve their marathon time from 3h27min38sec to 2h53min8sec by doubling endurance from $E_l = 3$ to $E_l = 6$ (corresponding to a change in the one-hour utilization from 79% to 87% of VO_{2max}), without any change in VO_{2max} or RE.
- We observed an initial linear increase of E_l with TRIMP, a plateau around $E_l = 7.5 \pm 2$ for $TRIMP \sim 25,000$, and a statistically significant final drop which may be due to over-training. This result suggests that there is an optimal TRIMP per TS, and the corresponding maximal endurance enables a close to optimal marathon race time for a given velocity v_m (see Fig. 3(a)).

A We have modified the results section to make our key findings stand out more clearly by adding subsections for each key finding. We could not add a itemized list of the key findings in the discussion section due to length restriction.

C (10) Minimize the use of acronyms where possible in the text to make it easier for readers to understand the paper. I suggest you remove the following:

- RS (racing season)
- TS (training season)
- RE (running economy)
- LDR (long distance runners?)

If the abbreviations are needed in a figure/table they are OK to use there, as long as they are defined in the caption.

A We have removed these acronyms.

C (11) by matching them with an universal, i.e., subject independent model: an universal \rightarrow a universal, a comma is needed after model

A done.

C (12) Our minimal model introduces effective parameters by measuring It is not clear what the authors mean by effective.

A We have removed "effective".

C (13) observations made by Hill in running world records: Reword to make it clear that it wasn't Hill who was running the world records :-).

A Thank you ;-). We have made this clear now.

C (14) Fig. 3 first shows a color coded plot of $T_{marathon}$ as function of the physiological parameters. This type of sentence is a better fit for a caption. In the Results it is preferable to describe specific findings. There are several instances of this in the Results.

A We have moved this type of sentences to the figure captions.

C (15) *To investigate the predictive power of our model in more detail, we applied our model also the RS with the marathon performance excluded: A word is missing from this sentence.*

A We have added the word "to" so it reads "... also to the race season ...".

C (16) *Consistent and inconsistent runners can be identified from the relative difference between our model estimates and actual race times. A better topic sentence (that covers the main focus of the paragraph) is needed to improve the logical flow of this section of the results. In general, a careful review of the entire paper to ensure each paragraph has a clear topic sentence would improve the quality of the manuscript.*

A We have reviewed and modified the manuscript to ensure that each paragraph has a clear topic and we added new subsections to the result section.

C (17) *Discussion, first paragraph: this should be broken into multiple paragraphs. The discussion of the limitations would be a natural split point.*

A done.

C (18) *This is an important advance over physiological testing in the laboratory where the required maximal effort is impossible to motivate for a distance of 20km or longer. I dont think the authors intend to mean that there is no use for lab-based testing. This is another place where the authors could soften their language. (e.g., important advance → important complement).*

In general, the primary point that stands out from the discussion is that the real-world data is a big improvement over lab testing. I dont think this is the most important point (as lab-based testing in a controlled environment still has great value). I would instead focus more on reviewing the specific new insights about running, training, and performance that were gleaned from the analysis.

A We agree. We have modified the manuscript in general to give a more balanced view of "real-world" data and lab testing, and focused more on the new insights from our analysis.

C (19) *Methods: Only TS with 30 or more runs were considered. What is the rationale for this choice? Was there any requirement from the minimum chronological length of the training season? Was there any sensitivity to these or other threshold choices discussed in the paragraph?*

A This minimum run condition for training season was applied so that runner had at least trained once per week on average during the 180 day long training season. Smaller number of runs could mean an interrupted training (e.g. due to injury), and hence relation to performance would be less reliable.

C (20) *Check for redundancy between material included in the Methods and Results.*

A We believe that the Methods section should be self-contained to allow a complete account of the applied procedures. However, we do have reduced some redundancy by combining some part of the Methods section with the appropriate paragraph of the Result section, please see next point.

C (21) *The following passage is a better fit for the results or discussion than the Methods. "For our two parameter model, the quality of the fitting could be probed for all RS with more than two races. For those RS we found a rather low average error of only 2.0% between the computed and actual race times. Another applicability test of our model is the estimation of the marathon finishing time from equation (1) when the parameters v_m and γ_1 are obtained from the RS without the marathon. Given all the possible uncertainties in marathon racing that are beyond the control of this study (e.g. weather, course profile, motivation of the athlete), the predictive power reflected by the results for marathon finishing time estimate in Fig. 4 is rather satisfying."*

A We have moved part of this passage to the Results section, please see lines 198 – 209 and 439 – 443.

Rebuttal letter**“Human running performance from real-world big data”
(NCOMMS-20-02292)**

Please find below our point-to-point answers to the reviewer comments (C: comment, A: answer). All changes in the manuscript are marked by colour highlighting (deleted text in red, newly added text in blue). Also, we have included line numbers in the manuscript (colour coded version) in order to make reference to changes in the point-by-point rebuttal letter.

Answer to Reviewer #3

We thank the reviewer for her/his time spent looking again over our manuscript and their comments that we address point-by-point in the following.

C I would like to see the reasoning that the author developed in replying to the reviewer’s comments more adequately integrated in the manuscript, especially in the discussion, and I hope the authors will show more consideration for the suggested references and comments. The respond is good, but it is not enough.

A As explained more clearly in the revised version, the aim of this work is neither a validation of our previously published model nor a comparison of our model to other existing models (which however are mentioned in our work). Rather, the aim of our work is to apply our model to real-world data and to extract performance parameters and relate them to racing performance and training. Due to this focus and due to length restrictions, we can not include all our reasoning from the previous reply in our manuscript. However, we have revised the manuscript overall to give a more balanced view of lab testing and our approach. To avoid confusion, we have changed the term “physiological parameters” to “performance indices”. In addition, we have added relevant references to previous work on theoretical concepts from exercise physiology in the Discussion section, please see lines 345 – 349.

Reviewers' Comments:

Reviewer #1:

Remarks to the Author:

I would like to thank the authors for the changes made to the this manuscript. From a conceptual perspective, I would say that I still disagree with the use of the MAP construct, as I think this is a flawed concept. However, I understand that it is commonly used and accepted by many, and that it serves the purpose of the present analysis. Aside from this comment (which is nothing but just a way of expressing my view), I am satisfied with the responses that the authors have provided and with the updated version of the manuscript. I think that focusing on performance rather than physiology makes this a much more solid and believable story. Thus, I have no further comments to make.